# Invariant Causal Mechanisms through Distribution Matching

## Abstract

Learning representations that capture the underlying data generating process is a key problem for data efficient and robust use of neural networks. One key property for robustness which the learned representation should capture and which recently received a lot of attention is described by the notion of invariance. In this work we provide a causal perspective and new algorithm for learning invariant representations. Empirically we show that this algorithm works well on a diverse set of tasks and in particular we observe state-of-the-art performance on domain generalization, where we are able to significantly boost the score of existing models.

## 1 Introduction

Learning structured representations which capture the underlying causal mechanisms generating data is of central importance for training robust machine learning models (Bengio et al., 2013; Schölkopf et al., 2021). One particular structure the learned representation should capture is invariance to changes in nuisance variables. For example, we may want the representation to be *invariant* to sensitive attributes such as the race or gender of an individual in order to avoid discrimination or biased decision making in a downstream task (Creager et al., 2019; Locatello et al., 2019; Träuble et al., 2021).

While learning invariant representations is thus highly important for fairness applications, it also appears in seemingly unrelated tasks such as domain adaptation (DA) and domain generalization (DG), where one aims to be invariant across the different domains (Muandet et al., 2013; Zemel et al., 2013; Ganin et al., 2016; Peters et al., 2016). For tasks such as DA and DG invariance across domains or environments implies to being invariant to the domain index, which thus is the "sensitive attribute" in this case and typically implies a change in the distribution of the data generating process. Being invariant to the domain index is thus a proxy to being invariant to latent unobserved factors that can change in distribution.

Established approaches for enforcing invariance in the learned representation usually aim to learn a representation whose statistical distribution is independent to the sensitive attribute e.g., by including an adversary during training (Ganin et al., 2016; Xie et al., 2017). As an adversary is essentially a parametric distributional distance, other approaches minimize different distribution distances, such as maximum mean discrepancy (MMD) (Louizos et al., 2017; Li et al., 2018b), or optimal transport (OT) based distances (Shen et al., 2018; Damodaran et al., 2018). To enforce independence, these methods add a regularizer to the loss that consists in the pairwise distributional distance between all possible combination of the sensitive attribute, i.e., $\text{dist}(p(z|d), p(z|d'))\forall d, d' \in D$. As such, the complexity of the loss grows quadratically in the size of the support of the sensitive attribute, which can limit the applicability of these models when the support of $D$ is large (Koh et al., 2021).

Despite the importance of learning invariant representations and their potential societal impact in the medical domain or fair decision making, most established approaches are still based on heuristics and specialized for different tasks at hand. We take first steps towards a unifying framework by viewing *invariance* as a property of a causal process (Pearl, 2009; Peters et al., 2017) and our key contributions can be summarized as follows:

- We introduce a unifying causal framework for invariant representation learning, which allows us to derive a new algorithm to enforce invariance through distribution matching.

One advantage of our algorithm is that only one distributional distance between two batches needs to be computed at each step, irrelevant of the size of the support of $D$.

- We define the notion of style variable and present some necessary and sufficient conditions under which being invariant to the domain index actually leads to invariance to the style variables. We argue that our proposal naturally captures most of the existing invariant representation learning tasks and datasets.
- Finally, we conduct a large number of experiments across different tasks and datasets, demonstrating the versatility of our framework. We obtain competitive results on the task of learning fair representations and we are able to significantly boost the performance of existing models using our proposed algorithm for the task of DG.

## 2 INVARIANT REPRESENTATION LEARNING ACROSS TASKS

In this section, we highlight how the learning of an invariant representation is a goal that is (implicitly) pursued in a large spectrum of machine learning tasks.

**Domain Adaptation**  The range of techniques used in Domain Adaptation and the different assumptions followed are vast (see Wilson and Cook (2020) for a more in depth review). Thus, we here concentrate only on a subset of the literature. A direction that is widely followed in DA, and which is the closest to our framework, is the alignment of the latent distribution of the source and target datasets. Under the *covariate shift* assumption, which assume that the labeling function $P(Y|X)$ is fixed, and that only $P(X)$ varies across environments, the goal is then to learn a representation $h(X)$ that is invariant across source and target and that remains useful to learn a discriminator on the source dataset. Ganin et al. (2016) uses a domain adversarial network to align the two latent spaces, whereas others uses distributional divergences directly, such as MMD (Baktashmotlagh et al., 2016), Wasserstein and optimal transport in general (Shen et al., 2018; Damodaran et al., 2018; Redko et al., 2017). DA under different assumptions, such as the case where both $P(Y)$ and $P(X|Y)$, have also been studied (Gong et al., 2016).

**Domain Generalization**  Though very similar to DA, DG differs in one significant way: the test domain is not observed at training time. As such, it is a way harder task as the test domain could exhibit arbitrary shifts in distribution, and the learned model is supposed to handle any *reasonable* shifts in distribution. Without any assumptions, there is little hope to obtain models that actually generalizes. Nevertheless, many inductive biases and models have been proposed, which have stronger assumptions than classical empirical risk minimization (ERM) (Vapnik, 1998).

Given its similarity to DA, similar models have been proposed, and most models work for both tasks. Nevertheless, until recently (Albuquerque et al., 2019; Deng et al., 2020), theoretical justification, e.g., for minimizing the distance between pairs of latent variables coming from different domains, was missing, as results from domain adaptation assumes that the test domain is observed. Without some assumptions, there exists no theoretical reasons to infer that a constant distribution of the latent variables across the training domains leads to better generalization on the test domains. Indeed, many benchmarks (Gulrajani and Lopez-Paz, 2020; Koh et al., 2021) show that it is difficult to create algorithms that consistently beat ERM across different tasks. Invariant representations for DG was first proposed by Muandet et al. (2013). This idea was then extended to use other distributional distances, such as MMD (Li et al., 2018b), Adversarial (Li et al., 2018d; Deng et al., 2020; Albuquerque et al., 2019), and Optimal Transport (Zhou et al., 2020) (see Table 1). On the theoretical side, both Albuquerque et al. (2019) and Deng et al. (2020) attempt to give theoretical grounding to the use of an adversarial loss by deriving bounds similar to what exists in DA.

**Domain Generalization and Causal Inference**  Many links between causal inference and domain generalization have been made, arguing that domain generalization is inherently a causal discovery task. In particular, causal inference can be seen as a form of distributional robustness (Meinshausen, 2018). In regression, one way of ensuring interventional robustness is by identifying the causal parents of $Y$, whose relation to $Y$ is stable. This can be achieved by finding a feature representation such that the optimal classifiers are approximately the same across domains (Peters et al., 2016; Rojas-Carulla et al., 2018; Arjovsky et al., 2019). Unfortunately, most of these models do not really apply to classification of structured data such as images, where the classification is predominantly

Table 1: Review of invariance across different tasks. Note that the general loss is defined as $\frac{1}{n}\sum_{i=0}^{n}\mathcal{L}(x_i, y_i) + \lambda \cdot \left(\text{dist}(z_1^n, z_{n+1}^N)\right)$.

| Task | Adversarial | MMD | Wasserstein |
|---|---|---|---|
| Distance equation | $\frac{1}{n}\sum_{i=0}^{n}\log\frac{1}{G_d(z_i)} + \frac{1}{n'}\sum_{i=n+1}^{N}\log\frac{1}{1-G_d(z_i)}$ | $\left\|\frac{1}{n}\sum_{i=0}^{n}\phi(z_i) - \frac{1}{n'}\sum_{i=n+1}^{N}\phi(z_i)\right\|_{\mathcal{H}}$ | $\frac{1}{n}\sum_{i=0}^{n}G_d(z_i) - \frac{1}{n'}\sum_{i=n+1}^{N}G_d(z_i)$ |
| Domain Adaptation | Ganin et al. (2016) Hoffman et al. (2017) | Baktashmotlagh et al. (2016) | Shen et al. (2018) Damodaran et al. (2018) |
| Domain Generalization | Ganin et al. (2016); Albuquerque et al. (2019) Li et al. (2018d;c); Deng et al. (2020) | Li et al. (2018b) | Zhou et al. (2020) |
| Fair Representation Learning | Edwards and Storkey (2015); Xie et al. (2017) Roy and Boddeti (2019) | Louizos et al. (2017) | Jiang et al. (2020) |

anti-causal and where the wanted invariance is not toward the pixels themselves but towards the unobserved generating factors. In a similar setting to ours, Heinze-Deml and Meinshausen (2021) tackles the task of image classification and propose a new model. A significant difference to our work is that they rely on the observation of individual instances across different views, i.e., the images are clustered by an ID.

**Fair Representation Learning**  Fair representation learning can also be viewed as an invariant representation learning task. This task consists in learning a representation that maximizes usefulness towards predicting a target variable, while minimizing information leakage of a sensitive attribute (e.g., gender, age, race). The seminal work in this field is Zemel et al. (2013), which aims at learning a multinomial random variable $Z$, with associated vectors $v_k$, such as the representation $Z$ is fair. More recent work directly learns a continuous variable $Z$ that has minimal information about the sensitive attribute, either through minimizing the MMD distance (Louizos et al., 2017), through adversarial training (Edwards and Storkey, 2015; Xie et al., 2017; Roy and Boddeti, 2019), or through a Wasserstein distance (Jiang et al., 2020).

## 3 INVARIANCE AS THE PROPERTY OF A CAUSAL PROCESS

In this section, we will first consider the assumptions for the causal process underlying the data generating mechanism using a structural causal model (SCM) type graph from Causality theory (Pearl, 2009) and following the causal view of learning disentangled representations (Suter et al., 2019), as illustrated in Figure 1.

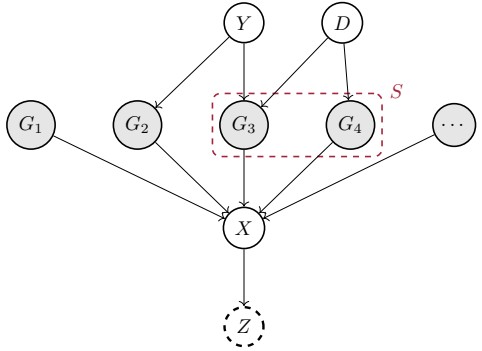

$G_1$ to $G_k$ represents all the factors of variation that generate the data, i.e., there exists a (one-to-one) function such that given all the factors, $X$ is fixed: $X \leftarrow g(G_1, \ldots, G_k)$

$Y$ is a target value that we may want to predict in a downstream task and is either known (supervised setting) or unobserved (unsupervised). $D$ is another confounder that we want to be invariant to. It can be a domain index, such as in DA and DG, or a sensitive attribute such as in fairness. We will assume for now that $D$ does not have an effect on $Y$.

Lastly, the generative factors $G_i$ are assumed to not have any causal relations between them, and any correlation between some factors may only come from a hidden confounder. This assumption is similar to the assumptions in Suter et al. (2019). Furthermore, in this work, we assume that the label $Y$ and $D$ directly have an effect on the latent generating factors. In this setting, $Y$ and $D$ are thus independent.

Figure 1: A direct acyclic graph (DAG) exhibiting our assumptions on the data generating process. We suppose that the data $X$ is a function of unobserved generative factors $G$ (e.g., background colors, brightness, noise, shape). There may exist some confounders $Y$ and $D$ that are parents of the generative variables. $Y$ is the variable that we want to predict. $D$ is the variable we want to be invariant to. Only $X$, $D$ and possibly $Y$ are observed at training time. The representation variable $Z$ is a function of the data $X$ that we *create* at training time.

Given our data generating framework, we can now give some definitions, especially the notion of style generating factors.

**Definition 3.1.** We call style variables the set of variables $G$ that are children of $D$ in the DAG. We denote this set $S$.

**Observation 3.1.** *$X$ and $Z$ are independent from $D$ given $S$, as they are $d$-separated from $D$ by the set $S$ in the graph.*

To the best of our knowledge there is no consistent and widely accepted definition of an invariant representation, yet. Using the above framework, we propose the following definition:

**Definition 3.2.** We say that a representation $Z$ is *invariant* to a variable $D$ if and only if $D$ has no total causal effect[1] on $Z$.

This definition of invariance is very robust since it guarantees that no intervention on the variable $D$ can break the independence between $Z$ and $D$. This is particularly relevant in application such as fair and private representation learning, as we may not want that intervening on the distribution of a sensitive variable breaks the property of fairness or privacy of a representation.

The goal of invariant representation learning can then be described as creating a new variable $Z = f(X)$ such that $D$ has no total causal effect on $Z$. In a way, we can view it as adding a new variable in the SCM and learning its structural equation. If we assume that our distribution follows our proposed SCM (Figure 1), then absence of total causal effect is equivalent to independence, as $D$ has no parent in the DAG. We use this assumption of $D$ having no causal parents in Theorem 3.1.

**Theorem 3.1.** Under the assumption of the graph in Figure 1, we have that:

$Z$ is independent from $D$ (equivalently, $D$ has no total causal effect on $Z$, or $p(z|d) = p(z|d')$ for all $d, d'$) $\iff p(z|do(d = N_d)) = p(z)$ for all $N_d$ (intervention on the distribution of $D$).

In summary, Theorem 3.1 states that no total causal effect is equivalent to independence given the right assumptions, and that having a constant marginal distribution of $Z$ under different mixtures of $D$ leads to independence. The proof is presented in Appendix B.

In the case where the full support of $D$ is observed during training, we have the guarantee that independence in training $I(Z; D) = 0$ will hold in any test setting. However, in DG for example, we observe a new value of $D$ at test time. Having $p(z|d_{train_i}) = p(z|d_{train_j}) \forall i, j$ does not guarantee that $p(z|d_{test}) = p(z|d_{train_i})$. Indeed, in this setting, the variable $D$ works more like an index, where each value indicates a domain where the distribution of $X$ has changed. Invariance to the variable $D$ is thus a proxy to being invariant to the unobserved style variables (see Definition 3.1).

## 4 AN ALGORITHM FOR INVARIANT LATENT VARIABLE DISTRIBUTIONS

**Necessary Condition for Invariant Representation Learning** We first start by studying whether invariance to $D$ may lead to invariance (or at least independence) to the style variables. For simplicity, suppose we are given two datasets drawn from two distributions $p_1$ and $p_2$ on $X$, i.e., $D \in \{1, 2\}$. We assume that only the style variables had their distribution changed across the two datasets.

The goal is to learn a representation of $X$ that is invariant to the style variables. We thus need to learn an encoder $f \in \mathcal{F}, z = f(x)$.

**Theorem 4.1.** Independence to $D$ is a necessary condition for the representation to be invariant to the style variables.

The proof is trivial and can be found in Appendix B. As this theorem shows, having a representation $z$ that is invariant to $D$ is at least necessary for invariance to the style variables. Unfortunately, it is in general not sufficient. It may be sufficient under some assumptions on the distribution of the style variables, but this is a direction we will not follow as it often leads to a very restrictive setting.

---

[1]A variable $i$ has a total causal effect on a variable $k$ if and only if: $X_i \not\perp\!\!\!\perp X_k$ in $P^{do(X_k = \tilde{N}_k)}$ for some random variable $\tilde{N}_k$ (Definition 6.12 in Peters et al. (2017)).

**Sufficient conditions for Invariant Representation Learning** We here present an idealistic setting where invariance to $D$ is sufficient. Obviously, this setting is almost never observed in practice. It nevertheless gives a sense of how difficult the task is and of what kind of direction we should follow.

**Theorem 4.2.** If we are given a (possibly infinite) number of domains, where each domain exhibits a different possible intervention on the style variables $S$, then independence to $D$ implies no total causal effect of $S$ on $Z$.

The proof can be found in Appendix B. We here take the view that the variable $D$ indexes an intervention, which is a common view in causal inference via DG (see Section 2). It also shows that there may always be a test domain where we observe an intervention to which our representation is not invariant, especially if it changes the support of the style variables. For examples, if we want to be invariant to the brightness in an image, we can only do so to a certain extent, as being invariant to full brightness (i.e., a white image) would force a constant and non-informative representation. This type of question is similar to what is studied in adversarial robustness. Consequently, we need assumptions on what types of interventions we may encounter in the wild. One reasonable assumption is that the full possible support of the style variables is observed during training. We then need to be invariant to any combination of possible values of the style variables. If the style variables are not observed, we cannot directly intervene on them. One possibility is thus to use the variable $D$ as a proxy to simulate interventions on the style variables.

**Conjecture 4.3.** Given a finite number of domains, we can create new domains via mixtures of the given domains, which simulates new types of interventions on the style variables $S$.

The idea is that having these new created domains may facilitate identifiability of a representation $Z$ invariant to the style variables $S$, as in a way it gets us closer to the conditions of Theorem 4.2.

**A New Algorithm for Invariant Representation Learning** Based on the underlying assumptions of Figure 1, the equivalence proved in Theorem 3.1 and Conjecture 4.3, we present a new algorithm to learn a representation invariant to interventions on $D$. This algorithm could be useful for example when we have a large number of different values of $D$, where enforcing an invariant $p(z|d)$ is hard to optimize (pairwise distances between distributions). Instead, we change the distribution of $D$ across batches (simulated intervention) and take the distribution distance between pairs of batches. It also follows Conjecture 4.3: as each time we create a batch with a different distribution of $D$, it is equivalent to drawing a batch from a created domain, which is a mixture of the initially given domains. We formulate the optimization goal as follows:

$$\min_{Z=f(X)} \mathcal{L}(Y, c(Z)), \quad \text{s.t. } p(Z) = \text{const } \forall N_d.$$

We relax this constraint by taking the dual formulation and by approximating the maximum distance between two possible interventions on $D$ by the average distance:

$$\min_{Z=f(X)} \mathcal{L}(Y, c(Z)) + \lambda \cdot \mathbb{E}_{N_d, N'_d} \left[ \text{dist}(p(Z|do(d = N_d)), \ p(Z|do(d = N'_d))) \right] \tag{1}$$

where dist is a distance between distributions (see Appendix A for possible distances), and $N_d, N'_d$ are interventions on the distribution of $d$.

First, this algorithm gives us a new method to learn invariance when our dataset follows the assumption of Figure 1. If the dataset indeed fulfills our assumptions, we theoretically have that the algorithm will work asymptotically. Unfortunately, these assumptions are not testable. There is thus two possible outcomes: either the optimization converges and we cannot reject that the dataset follows our assumptions, or it does not converge and it probably means that the causal relationships of the variables of our dataset are different than assumed.

Second, as stated in Conjecture 4.3, being invariant to different distributions of $D$ may lead to greater invariance to the style variables. This is especially useful in DG. In Section 5, we experimentally show that our algorithm is indeed a viable method to learn invariance, and that it can also be more

favorable in settings such as DG. We present a possible practical implementation of our algorithm 1 below.

---

**Algorithm 1:** Our algorithm for invariant representation learning.

---

1 Let $d$ be the number of domains;
2 Let $n > 0$ be the number of samples drawn from each class at each step;
3 **begin**
4      Draw a batch $b_i$ of $n$ samples for each domain;
5      $B1, B2 \leftarrow \emptyset, \emptyset$;
     `// We create two batches` $B1$ `and` $B2$ `that approximate the`
        `interventions` $N_d$ `and` $N_d'$ `of eq. (1)`
6      **for** $i \leftarrow 1$ to $d$ **do**
7         $s \sim \mathcal{U}(0, n)$;
8         $B1, B2 \leftarrow (B1, b_i[: s]), (B2, b_i[s :])$;
        `// Concatenate B1 and B2 with a slice of` $b_i$
9      **end**
10      $Z1, Z2 \leftarrow f(B1), f(B2)$;
11      $loss \leftarrow \text{dist}(Z1, Z2)$;
12      **return** $loss$;
13 **end**

---

## 5 EMPIRICAL EVALUATION

### 5.1 FAIR REPRESENTATION LEARNING

In this section, we present some experiments on Fair representation learning. Here, the goal is not necessarily to obtain better results compared to other baselines. More importantly, we want to show that: (i) Fair representation learning is also an invariant representation learning task, and it is covered by our unifying framework; (ii) Our algorithm is applicable to a wide range of tasks, as it also gives competitive results on this task; (iii) We can control the strength of invariance via the hyperparameter $\lambda$; (iv) Fair representation learning datasets probably also follow our proposed data generation graph.

In the context of fair representation learning, the variable $D$ we want to be invariant to here corresponds to what is usually referred to as the *sensitive* variable. A sensitive variable is a variable that should not have an effect on the predictions of a classifier or regressor. Some examples are the sex, the race or the age of an individual. If we can construct a representation that does not contain information about the sensitive variable, there is no way for a model built on top of this representation to base its prediction on the sensitive variable. Unfortunately, in most datasets, the sensitive variable is actually predictive for the target variable, i.e., the value we are trying to predict. This introduces a trade-off between fairness and accuracy of a model.

**Datasets** We run experiments on two datasets from the UCI ML-repository (Asuncion and Newman, 2007), the Adult and German dataset. The German dataset consist in predicting whether an individual has good or bad credit, while the sensitive attribute is the gender. The Adult dataset consists in predicting whether the annual income of an individual is more or less than $50, 000\$$, and the sensitive attribute is the gender. See Table 7 in the Appendix for a summary of the datasets. We also report the size of the majority class for the sensitive and target attribute for each dataset. A fair model should have a sensitive accuracy that is close or below the size of the majority sensitive class, while having a target accuracy as high as possible.

**Experiment Design** To run our experiments, we reuse the code from Roy and Boddeti (2019) and add our model. We also empirically modify the default latent representation size such that it is optimizable using the MMD distance. As for the synthetic experiment, after training of the encoder, we freeze it and learn two discriminators: one for the target and one for the sensitive attribute. The target discriminators is trained for 100 epochs and the adversary discriminator for 150 epochs. We report the best achieved test accuracy. The goal of this setup is to assess how much information can be extracted from the representation regarding the target and sensitive variables. For each value of $\lambda$ regularization, we run the experiment three times, and report mean and standard deviation. We

Table 2: Comparison to other existing models on the Adult dataset.

| Model | Target Accuracy | Adversary Accuracy |
|---|---|---|
| **CausIRL with MMD (ours)** | 85.0 | 69.8 |
| ML-ARL (Xie et al., 2017) | 84.4 | 67.7 |
| MaxEnt-ARL (Roy and Boddeti, 2019) | 84.6 | 65.5 |
| LFR (Zemel et al., 2013) | 82.3 | 67 |
| VFAE (Louizos et al., 2017) | 81.3 | 67 |

also report the best obtained model and compare it to other baselines. The goal is to show what performance we can potentially achieve with our algorithm and see how it compare with existing models.

### 5.1.1 ADULT DATASET

The encoder is a neural network with one hidden layer of size 7, and a latent representation size of 2. It is trained for 150 epochs using the Adam optimizer (Kingma and Ba, 2015), with learning rate of $1 \times 10^{-4}$ and weight decay of $5 \times 10^{-2}$. The discriminators are two-hidden-layer neural networks, with hidden layers of size 64 and 32. Both are optimized using Adam with learning rate of 0.001 and weights decay of 0.001. The learning rate of the discriminators is adjusted with Cosine Annealing. Train batch size is set to 128 and test batch size to 1000.

Results are summarized in Figure 2, as well as a comparison with other baselines in Table 2. The results are as we expected: stronger regularization leads to stronger invariance towards the sensitive attribute. We also get that there is a trade-off between target accuracy and adversary accuracy, as the sensitive attribute is informative towards predicting the target. Compared to other baselines, our best model performs well, as it has the best target accuracy for a slightly higher adversary accuracy.

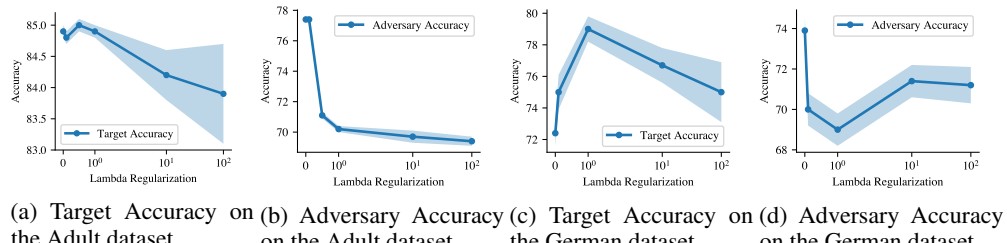

(a) Target Accuracy on the Adult dataset.
(b) Adversary Accuracy on the Adult dataset.
(c) Target Accuracy on the German dataset.
(d) Adversary Accuracy on the German dataset.

Figure 2: Target and adversary Accuracies for the Adult and German dataset for different strength of regularization.

### 5.1.2 GERMAN DATASET

The encoder is a neural network with two hidden layers of size 15 and 8, and a latent representation size of 32. It is trained for 150 epochs using the Adam optimizer, with learning rate of $1 \times 10^{-4}$ and weight decay of $5 \times 10^{-2}$. The discriminators are two-hidden-layer neural networks, with hidden layers of size 10. Both are optimized using Adam with learning rate of 0.001 and weights decay of 0.001. The learning rate of the discriminators is adjusted with Cosine Annealing. Train batch size is set to 64 and test batch size to 100.

Results are summarized in Figure 2, as well as a comparison with other baselines in Table 3. Here, we observe that 1.0 is a clear optimal value for $\lambda$, as it gives the highest target accuracy and the lowest adversary accuracy. A bit more surprisingly, we observe that higher regularization can give lesser invariance, which we can interpret as a form of over-regularization. Compared to other methods, we obtain competitive results as we get the smallest adversary accuracy, even below the majority prediction, while still obtaining the second best target accuracy.

Table 3: Comparison to other existing models on the German dataset.

| Model | Target Accuracy | Adversary Accuracy |
|---|---|---|
| **CausIRL with MMD (ours)** | 80.3 | 67.0 |
| ML-ARL (Xie et al., 2017) | 74.4 | 80.2 |
| MaxEnt-ARL (Roy and Boddeti, 2019) | 86.33 | 72.7 |
| LFR (Zemel et al., 2013) | 72.3 | 80.5 |
| VFAE (Louizos et al., 2017) | 72.7 | 79.7 |

## 5.2 Domain Generalization

**Datasets** For this experiments, we test on seven datasets: ColoredMNIST (Arjovsky et al., 2019), RotatedMNIST (Ghifary et al., 2015), VLCS (Fang et al., 2013), PACS (Li et al., 2017), OfficeHome (Venkateswara et al., 2017), TerraIncognita (Beery et al., 2018) and DomainNet (Peng et al., 2019). In the Appendix, Table 8 shows sample images for each dataset under different domains and Table 9 presents each dataset's characteristics.

**Experiment Design** We run our experiments with the DomainBed (Gulrajani and Lopez-Paz, 2020) testbed, which is a recent widely used testbed for DG. We choose this setup as it allows for a highly fair and unbiased comparison with other existing models. DomainBed was designed to be reproducible, to give each algorithm the same amount of hyperparameter search, and to accurately estimate the variance in performance. Three model selection methods are considered: training-domain validation (all training models are pooled and a fraction of each of them is used as validaiton set), leave-one domain-out cross-validation (cross validation is performed using a different domain as validation, and the best models is retrained on all training domains) and test-domain validation set (a fraction of the test domain is used as validation set). The first two methods are closer to a realistic setting, whereas oracle validation allows us to evaluate whether there exists headroom for improvement. Training-domain validation assumes that all training domains and the test domain follows a similar distribution, as we pool all the training domains during training. On the other hand, leave-one domain-out cross-validation is closer to our assumption, as it optimizes for generalization to an unseen domain that is assumed to follow a different distribution.

**Proposed Models** We take two existing models, MMD and CORAL, based on matching distribution across domains, and propose two new models, CausIRL with MMD and with CORAL. These two new algorithm simply consist in changing how the regularization loss is computed according to our proposed algorithm, i.e., instead of taking pairwise distances across domains, we compute distances between batches that follow different domain distributions. We thus want to see if this simple change in the algorithm leads to better performance, which may be due as we conjecture to greater invariance to the style variables, as well as the fact that it may be easier to optimize. The hyperparameter $\lambda$ is drawn randomly in $10^{\text{Uniform}(-1,1)}$.

### 5.2.1 Model Selection: Leave-One-Domain-Out Cross-Validation

We now look at the DG experiment results for the leave-one-domain-out cross-validation model selection method. The results are summarized in Table 4. The result for this model selection method are the most relevant as it closely follows our assumptions on the training and test distributions. The results for the other model selection methods can be found in Appendix E.

Here, the overall performance of CausIRL with CORAL is almost identical to CORAL. Nevertheless, there are some differences when looking at the performance on individual datasets. CausIRL with CORAL overperform CORAL on PACS, TerraIncognita and DomainNet, where CORAL performs better on VLCS and OfficeHome. However we should note that only the overperformance of CausIRL with CORAL over CORAL on DomainNet is statistically significant when looking at the confidence intervals of the average accuracies.

For CausIRL with MMD, we observe a significant boost in the overall performance compared to MMD. CausIRL with MMD performs better on almost all datasets, and we also observe a significant

Table 4: Domain Generalization experimental results for the leave-one-domain-out cross-validation model selection method.

| Algorithm | ColoredMNIST | RotatedMNIST | VLCS | PACS | OfficeHome | TerraIncognita | DomainNet | Avg |
|---|---|---|---|---|---|---|---|---|
| **CausIRL with CORAL (ours)** | $39.1 \pm 2.0$ | $97.8 \pm 0.1$ | $76.5 \pm 1.0$ | $83.6 \pm 1.2$ | $68.1 \pm 0.3$ | $47.4 \pm 0.5$ | $\mathbf{41.8 \pm 0.1}$ | 64.9 |
| CORAL | $39.7 \pm 2.8$ | $97.8 \pm 0.1$ | $\mathbf{78.7 \pm 0.4}$ | $82.6 \pm 0.5$ | $\mathbf{68.5 \pm 0.2}$ | $46.3 \pm 1.7$ | $41.1 \pm 0.1$ | **65.0** |
| **CausIRL with MMD (ours)** | $36.9 \pm 0.2$ | $97.6 \pm 0.1$ | $78.2 \pm 0.9$ | $\mathbf{84.0 \pm 0.9}$ | $65.1 \pm 0.7$ | $\mathbf{47.9 \pm 0.3}$ | $38.9 \pm 0.8$ | 64.1 |
| MMD | $36.8 \pm 0.1$ | $97.8 \pm 0.1$ | $77.3 \pm 0.5$ | $83.2 \pm 0.2$ | $60.2 \pm 5.2$ | $46.5 \pm 1.5$ | $23.4 \pm 9.5$ | 60.7 |
| ERM | $36.7 \pm 0.1$ | $97.7 \pm 0.0$ | $77.2 \pm 0.4$ | $83.0 \pm 0.7$ | $65.7 \pm 0.5$ | $41.4 \pm 1.4$ | $40.6 \pm 0.2$ | 63.2 |
| IRM | $40.3 \pm 4.2$ | $97.0 \pm 0.2$ | $76.3 \pm 0.6$ | $81.5 \pm 0.8$ | $64.3 \pm 1.5$ | $41.2 \pm 3.6$ | $33.5 \pm 3.0$ | 62.0 |
| GroupDRO | $36.8 \pm 0.1$ | $97.6 \pm 0.1$ | $77.9 \pm 0.5$ | $83.5 \pm 0.2$ | $65.2 \pm 0.2$ | $44.9 \pm 1.4$ | $33.0 \pm 0.3$ | 62.7 |
| DANN | $\mathbf{40.7 \pm 2.3}$ | $97.6 \pm 0.2$ | $76.9 \pm 0.4$ | $81.0 \pm 1.1$ | $64.9 \pm 1.2$ | $44.4 \pm 1.1$ | $38.2 \pm 0.2$ | 63.4 |
| CDANN | $39.1 \pm 4.4$ | $97.5 \pm 0.2$ | $77.5 \pm 0.2$ | $78.8 \pm 2.2$ | $64.3 \pm 1.7$ | $39.9 \pm 3.2$ | $38.0 \pm 0.1$ | 62.2 |
| VREx | $36.9 \pm 0.3$ | $93.6 \pm 3.4$ | $76.7 \pm 1.0$ | $81.3 \pm 0.9$ | $64.9 \pm 1.3$ | $37.3 \pm 3.0$ | $33.4 \pm 3.1$ | 60.6 |

Table 5: Performance results of our proposed datasets on Camelyon17 and RxRx1 compared to other baselines.

| Algorithm | Camelyon17 | RxRx1 |
|---|---|---|
| **CausIRL with CORAL (ours)** | $62.7 \pm 9.4$ | $29.0 \pm 0.2$ |
| CORAL | $59.5 \pm 7.7$ | $28.4 \pm 0.3$ |
| **CausIRL with MMD (ours)** | $63.4 \pm 11.2$ | $28.9 \pm 0.1$ |
| MMD | $64.6 \pm 10.5$ | $28.2 \pm 0.2$ |
| ERM | $\mathbf{70.3 \pm 6.4}$ | $\mathbf{29.9 \pm 0.4}$ |
| GroupDRO | $68.4 \pm 7.3$ | $23.0 \pm 0.3$ |
| IRM | $64.2 \pm 8.1$ | $9.9 \pm 1.4$ |

leap in performance on DomainNet going from 23.4% to 38.9%. This shows that the inductive bias of our algorithm matches the real inductive bias of DG far more accurately.

# 6 REAL-WORLD DOMAIN GENERALIZATION

In this section, we run experiments on more realistic distributional shifts. We use the Wilds (Koh et al., 2021) benchmark and run experiments on two datasets: Camelyon17 (Bandi et al., 2018) and RxRx1 (Taylor et al., 2019). Camelyon17 consists in predicting whether a region of tissue contains tumor tissue, while being invariant to the hospitals where the sample was taken. The goal is to obtain a model that generalizes across hospitals, as hospital specific artifacts of the data collection process can vary. RxRx1 consists of cell images, where the cells received some genetic treatment (as well as no treatment). The goal is to predict the genetic treatment among $1,139$ possible treatments. Here, we want to be invariant to the *batch* the cells come from, as it is a common observation that batch effect can greatly alter the results.

We test our two proposed models, CausIRL with CORAL and with MMD on both datasets. For the RxRx1 dataset, we use the same hyperparameters than for the CORAL model in the Wilds implementation. For Camelyon17, we change the number of group per batch to three and the batch size to 60. The results are summarized in Table 5. As for the DG experiments on DomainBed before, we observe that CausIRL with CORAL performs better than CORAL. Moreover, CausIRL with MMD performs slightly better than CausIRL with CORAL on Camelyon17 and similarly on RxRx1. Unfortunately, all models perform worse than simple ERM. Nevertheless, we again observe that our proposed models work competitively even on a realistic dataset, and that our proposed algorithm to compute the distributional distance regularization is better than how it is usually done.

# 7 CONCLUSION AND FUTURE WORK

In this work, we provided a causal perspective on invariant representation learning. We defined style variables in the context of our framework and developed theory on what conditions are necessary or sufficient to be invariant towards the style variables. Based on these theoretic insights and the assumptions on the data generating process, we then proposed a new algorithm for enforcing invariance to style variables in the learned representations. We empirically demonstrated that our algorithm is versatile as it works on a diverse set of tasks and datasets. In particular, it performs strongly in DG, where we obtain state-of-the-art performance.

## 8 REPRODUCIBILITY STATEMENT

Regarding the theoretical results, all results are based on the assumptions of Figure 1, and the proofs can be found in Appendix B. For the fair representation learning experiments, the architectures and training procedures are precisely described in Section 5.1, and we reused the code of Roy and Boddeti (2019) to ensure a fair comparison to existing models. For the DG experiments, we used two benchmarks, DomainBed (Gulrajani and Lopez-Paz, 2020) and Wilds (Koh et al., 2021), which were designed to be reproducible and unbiased. The hyperparameter used can be found in the main text. Regarding the implementation of our algorithm, the pseudo-code in algorithm 1 and the python code snippets in the Appendix should be sufficient to accurately implement our proposed models. For all experiments, our implementation of the MMD and CORAL distance are taken from the DomainBed (Gulrajani and Lopez-Paz, 2020) code. Our code implementations are provided with all the necessary details in the corresponding sections in the appendix. For the implementation details for fair representation learning we refer to the detailed documentation of the code in section D and for domain generalization we refer to section E.

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

# A  BACKGROUND

In this chapter, we review some necessary theoretical background that is used for our theoretical results, new proposed method as well as previous works. We review the main elements of Causality theory, which is the central theoretical basis used to model the problems we study and to design new methods and algorithms. We also review some distributional distances used in this work, as our main goal is to study invariant representation learning via invariant latent variable distribution.

## A.1  CAUSALITY

Causality essentially is the study of cause and effects, which goes beyond the study of statistical associations from observational data. This allows to reason about the notion of *interventions*, such as a treatment in medicine. The expected effect of an intervention is in general not equivalent to statistical conditioning, which calls for a more profound understating of the data generating process that goes beyond correlations between variables. We here focus on Pearl's view of causality Pearl (2009), which mainly relies on DAGs.

A DAG allows to represent the relations between variables, where each variable is represented by a node in the graph. Consequently, we can interpret directed edges between nodes as the existence of a causal effect from the parent node (the cause) on the child node (the effect).

Let $G = (V, E)$ be a DAG, and $P$ be a distribution. We say that $(G, P)$ is a causal DAG model if for any $W \subset V$, we have:

$$p(x_V | do(X_W = x'_W)) = \prod_{i \in V \setminus W} p(x_i | x_{pa_i}) \mathrm{I}(x_W = x'_W)$$

where $x_{pa_i}$ are the parents of node $i$ in graph $G$, I is the indicator function and $do(X_W = x'_W)$ denotes the intervention on the variables $X_W$. As we can see above, one of the properties of a causal DAG model is that the distribution factorizes according to the parents in the associated graph $G$.

### A.1.1  STRUCTURAL CAUSAL MODELS

A SCM can be seen as a more expressive version of a causal DAG model. Formally, an SCM consists of a collection $S$ of $d$ structural assignments, one per variable:

$$X_j \leftarrow f_j(X_{pa_j}, N_j)$$

where $X_{pa_j} \in X \setminus X_j$, and $N_1$ to $N_d$ are called the *noise variables* (Definition 6.2 of Peters et al. (2017)). The noise variables are assumed to be jointly independent.

For causal DAG models, we defined an intervention by $p(X_V | do(X_W = x'_W))$ (sometimes also written as $p^{do(X_W = x'_W)}(X_V)$), where the value of some variables are set to a constant value. With SCMs, we can give a more precise and general definition of interventions. An intervention now consists in replacing a subset of the collection $S$ of structural assignments by new functions. An intervention can thus consist in replacing a variable by a constant, a new random variable or even by changing the function and its arguments (i.e., its parents). The new distribution over the variables entailed by the new intervened SCM is denoted by $P^{do\left(X_k = \tilde{f}(X_{\widetilde{pa}_k}, \tilde{N}_k)\right)}$ (see Definition 6.8 of Peters et al. (2017) for more details).

With this definition, we are now equipped to reason about interventions, causes and effects. One important notion is the notion of *Total Causal Effect*.

**Definition A.1.** (Definition 6.12 in Peters et al. (2017)) We say that a variable $i$ has a total causal effect on a variable $k$ if and only if:

$$X_i \not\perp\!\!\!\perp X_k \text{ in } P^{do\left(X_k = \tilde{N}_k\right)}$$

for some random variable $\tilde{N}_k$.

A total causal effect between a variable $X_k$ and $X_i$ may only exist if there is a directed path from $i$ to $k$ in the DAG associated to our SCM. On the other hand, there may be no total causal effect between two variable even though there exists a directed path between them in the graph.

## A.2 DISTRIBUTIONAL DISTANCES

The main goal of this work is to study how invariance can be enforced by regularizing different latent spaces to have the same distribution. To this end, we thus need a differentiable distance or divergence between distributions that can be minimized during training. We here present the most commonly used distances in the literature.

### A.2.1 ADVERSARIAL

Adversarial training was first introduced in Goodfellow et al. (2014) as a new method for Generative modeling. Based on game theory, it can intuitively be described as a two player game, where each player is parameterized by a neural network. The Generator is a function that maps its input distribution to an output distribution. We call it the generated distribution and denote it by $p_g$. On the other hand, a Discriminator tries to distinguish between samples coming from the target dataset and samples produced by the Generator. At convergence, the Generator produces data that is distributed similarly to the target distribution, and thus it becomes impossible for the Discriminator to distinguish samples.

Formally, the objective of the two-player minimax game reads:

$$\min_G \max_D V(D, G) = \mathbb{E}_{\mathbf{x} \sim p_{data}(\mathbf{x})} \left[ \log D(\mathbf{x}) \right] + \mathbb{E}_{\mathbf{z} \sim p_{\mathbf{z}}(\mathbf{z})} \left[ \log \left( 1 - D(G(\mathbf{z})) \right) \right] \tag{2}$$

where $\mathbf{z}$ is the input, $\mathbf{x}$ comes from the target distributions, and the Discriminator $D$ should output $1$ when its input is a samples from the target, and $0$ otherwise. If the Discriminator is optimal for a given $G$, Equation 2 can be rewritten to show that the Generator actually minimizes the Jensen–Shannon divergence (JSD) between the generated and target distribution.

$$JSD\left(P \| Q\right) = \frac{1}{2} D_{\mathrm{KL}} \left( P \left\| \frac{1}{2} \left(P + Q\right) \right.\right) + \frac{1}{2} D_{\mathrm{KL}} \left( Q \left\| \frac{1}{2} \left(P + Q\right) \right.\right),$$

where $D_{\mathrm{KL}}$ is the Kullback-Leibler (KLd) divergence. It also can be shown that if both networks have sufficient capacity, and if the Discriminator is trained to optimality after each optimization step of the Generator, then the distribution of the Generator converges to the target distribution.

Adversarial training can thus be seen as a proxy distributional distance, which corresponds to the JSD at convergence. This concept of adversarial training has been extended to be used as a regularizer for latent spaces. It can for example be used to enforce a prior distribution on the latent space Makhzani et al. (2015). It can also be used to enforce two latent spaces to have the same distribution. Its use is often justified as wanting two latent spaces to seem *indistinguishable* for an adversary, which is supposed to force the encoder to discard what is not constant across the two input distribution. We argue that adversarial training is theoretically equivalent to minimizing any distributional divergence, and that only their optimization properties differentiate them. We will also later clarify the intuition of trying to discard the *idiosyncratic in favor of the universal*, and what it actually corresponds to when we look at the data generation process of a given dataset.

### A.2.2 MAXIMUM MEAN DISCREPANCY

MMD Gretton et al. (2006) is a distance based on empirical samples from two distributions, based on the distance between the means of the two sets of samples mapped into a reproducing kernel Hilbert space (RKHS). Let $\{X\} \sim P$ and $\{X'\} \sim Q$. Then, we have:

$$MMD(X, X')^2 = \left\| \frac{1}{n} \sum_{i=1}^{n} \phi(x_i) - \frac{1}{n'} \sum_{i=1}^{n'} \phi(x_i') \right\|$$

$$= \frac{1}{n^2} \sum_{i,j=1}^{n} k(x_i, x_j) + \frac{1}{n'^2} \sum_{i,j=1}^{n'} k(x_i', x_j') - \frac{2}{n \cdot n'} \sum_{i=1}^{n} \sum_{j=1}^{n'} k(x_i, x_j'),$$

where $k(\cdot, \cdot)$ is the associated kernel. One commonly used kernel is the Gaussian kernel $k(x, x') = e^{-\lambda \|x - x'\|^2}$. Asymptotically, for a universal kernel such as the Gaussian kernel, $MMD(X, X') = 0$ if and only if $P = Q$. Minimizing the MMD distance during training can thus be used to align two distributions.

### A.2.3 OPTIMAL TRANSPORT

OT has recently gained traction in machine learning research. It is particularly interesting as it defines multiple distances between distributions.

We present the main formulations of the OT problem and some distances following the notations of Peyré and Cuturi (2019). We concentrate on distances defined on discrete distributions.

Originally, OT can be described as the problem of transporting the mass from a set of points $\{x_1, \ldots, x_n\}$ to a set of destination points $\{y_1, \ldots, y_m\}$ that have finite capacities. Furthermore, the cost of transporting from a point to another is fixed by a cost matrix. There is obviously a natural analogy to logistics and planning. The OT problem thus is to fulfill this mass transport while minimizing the cost. To recast this as a distance between distribution, the mass of the source points is now described by an histogram $\mathbf{a} \in \{_+^n \colon \sum_i \mathbf{a}_i = 1\}$ and the destination capacities as an histogram $\mathbf{b} \in \{_+^m \colon \sum_i \mathbf{b}_i = 1\}$. The cost matrix is $\mathbf{C} \in _+^{n \times m}$ where $\mathbf{C}_{ij}$ describe the cost of transport from $x_i$ to $y_j$. The Kantorovich formulation of this problem is:

$$\mathrm{L}_{\mathbf{C}}(\mathbf{a}, \mathbf{b}) := \min_{T \in \mathbf{U}(\mathbf{a}, \mathbf{b})} \langle \mathbf{C}, T \rangle := \sum_{i,j} \mathbf{C}_{i,j} T_{i,j} \tag{3}$$

where

$$\mathbf{U}(\mathbf{a}, \mathbf{b}) := \left\{ T \in _+^{n \times m} \colon T \mathbb{1}_m = \mathbf{a}, T^T \mathbb{1}_n = \mathbf{b} \right\} \tag{4}$$

$T_{i,j}$ describes how much mass flows from $x_i$ to $y_j$. The constrain imposes that all the mass leaves $\mathbf{a}$ and that $\mathbf{b}$ is *filled*. Given the cost matrix, we hence can compute the divergence between two distributions, and this divergence elegantly comes with an exact description (the coupling matrix $T$) of how to go from the first configuration to the other. Furthermore, if $x_i, y_i \in \mathcal{X}$ lie in the same metric space with distance $d$ and the cost matrix satisfies $\mathbf{C}_{i,j} = d^p(x_i, y_j)$ for some $p \geq 1$, then:

$$\mathrm{W}_p(\mathbf{a}, \mathbf{b}) := \mathrm{L}_{\mathbf{C}}(\mathbf{a}, \mathbf{b})^{1/p} \tag{5}$$

is a *distance*, called the p-Wasserstein distance. This distance has had a large range of applications, notably for problems related to invariant representation learning (see Section 2).

## B PROOFS OF THEOREMS

**Theorem 3.1.** Under the assumption of the graph in Figure 1, we have that:

$Z$ is independent from $D$ (equivalently, $D$ has no total causal effect on $Z$, or $p(z|d) = p(z|d')$ for all $d, d'$) $\iff$ $p(z|do(d = N_d)) = p(z)$ for all $N_d$ (intervention on the distribution of $D$).

*Proof.* $\implies$ As $z$ is a descendant of $d$, the mechanism $p(z|d)$ is invariant to the distribution of $d$.

$$p^{do(d=N_d)}(z) = \int p^{do(d=N_d)}(z, d') \mathrm{d}d' = \int p^{do(d=N_d)}(z|d') N_d(d') \mathrm{d}d' = p(z) \int N_d(d') \mathrm{d}d'$$

$$= p(z)$$

$\impliedby$ As $p(z)$ is constant for all distribution of $d$, then it is also constant for deterministic distributions, i.e $\delta_d$.

$$p(z) = \int p(z|d')\delta_d(d')\mathrm{d}d' = p(z|d)$$

This holds for all values of $d$, which implies that $I(Z; D) = 0$. $\qquad\square$

**Theorem 4.1.** Independence to $D$ is a necessary condition for the representation to be invariant to the style variables.

*Proof.*
$$
\begin{aligned}
p(z|d=1) - p(z|d=2) &= \int p(z|S, d=1)[p(S|d=1)dS - \int p(z|S, d=2)p(S|d=2)]dS \\
&= \int p(z|S)[p(S|d=1) - p(S|d=2)]dS \\
&= p(z) \int [p(S|d=1) - p(S|d=2)]dS \\
&= 0
\end{aligned}
$$

$\qquad\square$

**Theorem 4.2.** If we are given a (possibly infinite) number of domains, where each domain exhibits a different possible intervention on the style variables $S$, then independence to $D$ implies no total causal effect of $S$ on $Z$.

*Proof.* From the definition of total causal effect, let's suppose by contradiction that there exists an intervention on $S$ such that $p^{do(S=\tilde{N}_s)}(z) \neq p(z)$. Let $\tilde{d}$ denote the domain that correspond to this intervention. We then have a value of $D$ such that $p(z|\tilde{d}) \neq p(z)$, which is a contradiction to $Z$ being independent to $D$. $\qquad\square$

## C   SYNTHETIC EXPERIMENT

We here implement a simple synthetic experiment to verify that our algorithm effectively enforces invariance to $D$ in a setting that exactly follows our assumptions. We also simplify the setting by considering that we directly observe the generative factors.

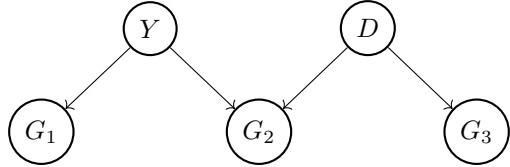

Figure 3: Causal DAG associated to our synthetic distribution.

The distribution is generated by the following set of structural equations:

$$
\begin{aligned}
Y &\leftarrow N_y; \\
D &\leftarrow N_d; \\
G_1 &\leftarrow Y + N_{G_1}; \\
G_2 &\leftarrow 2 \cdot Y + 2 \cdot D + N_{G_2}; \\
G_3 &\leftarrow D + N_{G_3};
\end{aligned}
$$

where $N_y$ and $N_d \sim Ber(0.5)$, $N_{G_i} \sim \mathcal{N}(0, 1)$. We draw the associated causal DAG in Figure 3. To create a dataset, we draw 1000 samples from our synthetic distribution and use 200 of them as test samples.

Table 6: Results of predictive accuracy for $Y$ (Target Accuracy) and $D$ (Adversary Accuracy) for different strength of regularization on our synthetic dataset.

| Regularization | 0.0 | 0.1 | 0.5 | 1.0 | 5.0 | 10.0 |
|---|---|---|---|---|---|---|
| Target Accuracy | $86.0 \pm 0.2$ | $86.2 \pm 0.5$ | $86.8 \pm 0.1$ | $86.7 \pm 0.3$ | $52.0 \pm 3.2$ | $54.2 \pm 5.1$ |
| Adversary Accuracy | $71.8 \pm 0.8$ | $63.8 \pm 0.9$ | $63.5 \pm 1.2$ | $60.5 \pm 0.2$ | $59.8 \pm 0.4$ | $51.0 \pm 0.0$ |

We then learn a representation that is invariant to $D$ and that is predictive towards $Y$ using our proposed loss. As a distributional distance, we use the MMD loss with Gaussian kernel. The architecture of the encoder is a neural network with one hidden layer of size 10 and a representation size of 5. The hidden layer is followed by a batch normalization and a ReLU activation. We use a batch size of 64 and train with the Adam optimizer (Kingma and Ba, 2015) for 200 epochs, with a learning rate of 0.001 and weight decay of $5 \times 10^{-5}$.

After training of the encoder, we freeze it and train two one layer linear discriminators: one to predict $Y$ and one to predict $D$. For each discriminators, we report the best achieved test accuracy. We run this experiment three times for each value of lambda regularization , $\lambda \in \{0.0, 0.1, 0.5, 1.0, 5.0, 10.0\}$. The results are summarized in Table 6 and Figure 4.

As expected, we can observe a strong correlation between the strength of regularization and the strength of invariance. We achieved perfect invariance with $\lambda = 10.0$, where the adversary accuracy is $50\%$, but target accuracy is only $54.2\%$. This is expected: as $Y$ and $D$ are strongly correlated, removing information on $D$ in the representation also reduces the predictive power of the representation. There thus is a trade-off between performance and invariance, that can be controlled via the value of $\lambda$. Finally, this experiment confirms that our proposed algorithm is a viable new method to learn invariance.

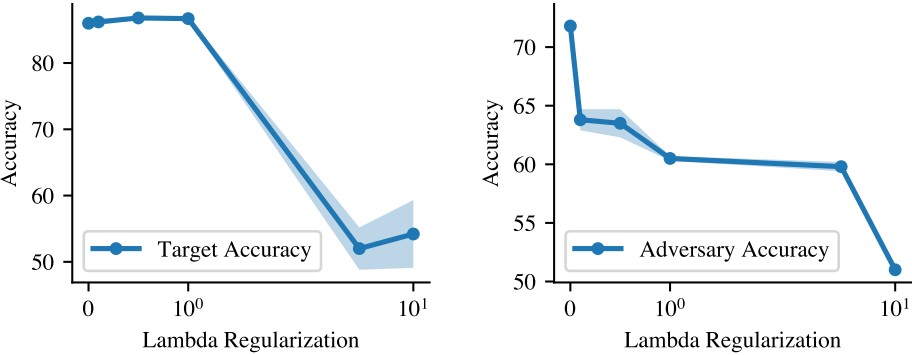

Figure 4: Graphical visualization of our results on the synthetic dataset. The values here are the same than in Table 6.

## D   FAIR REPRESENTATION LEARNING EXPERIMENT SUPPLEMENTS

**Compute Resources**   We run the experiements on NVIDIA GEFORCE RTX 2080 TI GPUs.

**Implementation**   Here is the implementation of our model, which is a class that we added to the code[2] of Roy and Boddeti (2019) to the `train.py` file:

```
class CausIRL_MMD:
    def __init__(self, data,
                 train_loader=None,
                 test_loader=None,
```

---

[2]https://github.com/human-analysis/MaxEnt-ARL

```python
            total_epoch=200,
            alpha=0.1,
            epsilon=0.1,
            use_cuda=False,
            resume=False,
            ckpt_filename=None,
            resume_filename=None,
            privacy_flag=True,
            privacy_option='maxent-arl',
            print_interval_train=10,
            print_interval_test=10
            ):
    # data info
    self.data = data
    self.train_loader = train_loader
    self.test_loader = test_loader
    self.n_sensitive_class = self.data.n_sensitive_class
    self.n_target_class = self.data.n_target_class

    # models
    self.adv_net = data.adversary_net
    self.target_net = data.target_net
    self.discriminator_net = data.discriminator_net

    # optimizer
    self.optimizer = data.optimizer
    self.discriminator_optimizer = data.discriminator_optimizer
    self.adv_optimizer = data.adv_optimizer
    self.target_optimizer = data.target_optimizer

    def my_cdist(x1, x2):
        x1_norm = x1.pow(2).sum(dim=-1, keepdim=True)
        x2_norm = x2.pow(2).sum(dim=-1, keepdim=True)
        res = torch.addmm(x2_norm.transpose(-2, -1),
                    x1,
                    x2.transpose(-2, -1), alpha=-2).add_(x1_norm)
        return res.clamp_min_(1e-30)

    def gaussian_kernel(x, y, gamma=[0.001, 0.01, 0.1, 1, 10, 100,
                                        1000]):
        D = my_cdist(x, y)
        K = torch.zeros_like(D)

        for g in gamma:
            K.add_(torch.exp(D.mul(-g)))

        return K

    def mmd(x, y):
        Kxx = gaussian_kernel(x, x).mean()
        Kyy = gaussian_kernel(y, y).mean()
        Kxy = gaussian_kernel(x, y).mean()
        return Kxx + Kyy - 2 * Kxy

    # loss
    self.kl_loss = nn.KLDivLoss()
    self.cross_entropy_loss = nn.CrossEntropyLoss()
    self.entropy_loss = EntropyLoss()
    self.nll_loss = nn.NLLLoss()
```

```python
        self.mse_loss = nn.MSELoss()
        self.mmd_loss = mmd

        # filename
        self.log_file_name = ckpt_filename+"_log.txt"
        self.adv_log_file_name = ckpt_filename+"_adv_log.txt"
        self.target_log_file_name = ckpt_filename + "_target_log.txt"
        self.checkpoint_filename = ckpt_filename
        self.adv_checkpoint_filename = ckpt_filename+"_adv.ckpt"
        self.target_checkpoint_filename = ckpt_filename + "_target.ckpt"

        # algorithm and visualization parameters
        self.alpha = torch.tensor([alpha*1.0], requires_grad=True)
        self.resume = resume
        self.epoch = 0
        self.gamma_param = 0.01
        self.plot_interval = 10
        self.print_interval_train = print_interval_train
        self.print_interval_test = print_interval_test
        self.use_cuda = use_cuda
        self.privacy_flag = privacy_flag
        self.privacy_option = privacy_option

        # local variables
        self.uniform = torch.tensor(1 / (self.data.n_sensitive_class))
        .repeat(self.data.n_sensitive_class)
        self.target_label = torch.zeros(0, dtype=torch.long)
        self.sensitive_label = torch.zeros(0, dtype=torch.long)
        self.sensitive_label_onehot = torch.FloatTensor(0,
        self.data.n_sensitive_class)
        self.target_label_onehot = torch.FloatTensor(0,
        self.data.n_target_class)
        self.inputs = torch.zeros(0, 0, 0)
        self.inputs.requires_grad = False
        self.batch_uniform = torch.FloatTensor(0, self.data.n_sensitive_class)
        self.epsilon = torch.tensor([epsilon]).float()

        if resume:
            assert os.path.isdir('checkpoint'), 'Error:␣no␣checkpoint␣directory␣foun
            if self.use_cuda:
                checkpoint = torch.load(os.path.join('checkpoint/',
                resume_filename))
            else:
                checkpoint = torch.load(os.path.join('checkpoint/',resume_filename),
                map_location=lambda storage, loc: storage)
            self.net = checkpoint['net']
            self.best_acc = 0  # checkpoint['acc']
            self.start_epoch = 0  # checkpoint['epoch']
            self.total_epoch = total_epoch  # + self.start_epoch

            for param in self.net.parameters():
                param.requires_grad = True
        else:
            self.net = data.net
            self.best_acc = 0
            self.start_epoch = 0
            self.total_epoch = total_epoch

        if self.use_cuda:
```

```python
            self.net = self.net.cuda()
            self.discriminator_net = self.discriminator_net.cuda()
            self.adv_net = self.adv_net.cuda()
            self.target_net = self.target_net.cuda()
            self.net = nn.DataParallel(self.net, device_ids=
            range(torch.cuda.device_count()))
            self.target_net = nn.DataParallel(self.target_net,
            device_ids=range(torch.cuda.device_count()))
            self.discriminator_net = nn.DataParallel(self.discriminator_net,
            device_ids=range(torch.cuda.device_count()))
            self.adv_net = nn.DataParallel(self.adv_net,
            device_ids=range(torch.cuda.device_count()))
            cudnn.benchmark = True
            self.inputs = self.inputs.cuda()
            self.target_label = self.target_label.cuda()
            self.sensitive_label = self.sensitive_label.cuda()
            self.sensitive_label_onehot = self.sensitive_label_onehot.cuda()
            self.target_label_onehot = self.target_label_onehot.cuda()
            self.uniform = self.uniform.cuda()
            self.batch_uniform = self.batch_uniform.cuda()
            self.alpha = self.alpha.cuda()

        self.best_loss = 1e16
        self.adv_best_acc = 0
        self.target_best_acc = 0
        self.t_losses, self.t_top1, self.d_losses, self.d_top1 =
        AverageMeter(), AverageMeter(), AverageMeter(), AverageMeter()
        self.e_losses, self.losses = AverageMeter(), AverageMeter()
        self.t_top5, self.d_top5 = AverageMeter(), AverageMeter()
        self.adv_losses, self.adv_top1, self.adv_top5,
        self.entropy_losses = AverageMeter(),
        AverageMeter(), AverageMeter(), AverageMeter()
        self.target_losses, self.target_top1, self.target_top5, self.target_entropy_
        AverageMeter(), AverageMeter(), AverageMeter(), AverageMeter()

    def perform_epoch(self, epoch, test_flag=False):
        if test_flag:
            self.net.eval()
            self.discriminator_net.eval()
            self.target_net.eval()
            loader = self.test_loader
            string = "Test"
            print_interval = self.print_interval_test
            data_size = len(self.test_loader)
        else:
            self.net.train()
            self.discriminator_net.train()
            self.target_net.train()
            loader = self.train_loader
            string = "Train"
            print_interval = self.print_interval_train
            data_size = len(self.train_loader)

        iteration = 0

        self.t_losses.reset()
        self.e_losses.reset()
        self.losses.reset()
        self.d_losses.reset()
```

```python
        self.t_top1.reset()
        self.d_top1.reset()
        self.t_top5.reset()
        self.d_top5.reset()
        self.entropy_losses.reset()

        for batch_idx, (inputs, target_label, sensitive_label) in enumerate(loader):

            batch_size = inputs.size(0)
            iteration += 1

            self.inputs.resize_(inputs.size()).copy_(inputs)
            self.target_label.resize_(target_label.size()).
            copy_(target_label)
            self.sensitive_label.resize_(
            sensitive_label.size()).copy_(sensitive_label)
            self.sensitive_label_onehot.resize_([batch_size,
            self.data.n_sensitive_class])
            self.sensitive_label_onehot.zero_()
            self.sensitive_label_onehot.scatter_(1,
            torch.unsqueeze(self.sensitive_label, 1), 1)
            self.target_label_onehot.resize_([batch_size,
            self.data.n_target_class])
            self.target_label_onehot.zero_()
            self.target_label_onehot.scatter_(1,
            torch.unsqueeze(self.target_label, 1), 1)
            self.batch_uniform.resize_([batch_size, self.data.n_sensitive_class])
            self.batch_uniform[:, :] = 1.0/(self.data.n_sensitive_class)
            self.batch_uniform.scatter_(1,
            torch.unsqueeze(self.sensitive_label, 1), 0)
            self.optimizer.zero_grad()

            _, z, e_prob = self.net(self.inputs)
            target_outputs, _, t_prob = self.target_net(z)
            t_loss = torch.nan_to_num(self.cross_entropy_loss(
            target_outputs+1e-16, self.target_label))
            entropy_loss = torch.tensor(0)
            s_loss = torch.tensor(0)

            if self.privacy_flag:
                #d_outputs, _, d_prob = self.discriminator_net(z)
                #entropy_loss = -self.entropy_loss(d_prob)
                first = None
                second = None
                for i in range(self.n_sensitive_class):
                    ind = self.sensitive_label == i
                    z_ = z[ind]
                    slice = random.randint(0, len(z_))
                    if first is None:
                        first = z_[:slice]
                        second = z_[slice:]
                    else:
                        first = torch.cat((first, z_[:slice]), 0)
                        second = torch.cat((second, z_[slice:]), 0)
                if len(first) > 1 and len(second) > 1:
                    s_loss = torch.nan_to_num(self.mmd_loss(first, second))

                loss = t_loss + self.alpha*s_loss
            else:
```

```python
            loss = t_loss

        if not test_flag:  # update weights
            self.optimizer.zero_grad()
            self.target_optimizer.zero_grad()
            loss.backward()
            self.optimizer.step()
            self.target_optimizer.step()

        # measure accuracy and record loss for learner
        t_prec1 = accuracy(t_prob.data, self.target_label.data)
        t_prec5 = accuracy(t_prob.data, self.target_label.data,
        topk=(int(np.min([5, self.n_target_class])),))
        self.t_losses.update(t_loss.data.item(), batch_size)
        self.e_losses.update(s_loss.data.item(), batch_size)
        self.losses.update(loss.data.item(), batch_size)
        self.t_top1.update(t_prec1[0], batch_size)
        self.t_top5.update(t_prec5[0], batch_size)
        self.entropy_losses.update(s_loss.data.item(), batch_size)

        if self.privacy_flag:
            if not test_flag:
                self.discriminator_net.train()
            d_outputs, _, a_prob = self.discriminator_net(z.detach())
            d_loss = self.nll_loss(torch.log(a_prob+1e-16),
            self.sensitive_label)

            if not test_flag:
                self.discriminator_optimizer.zero_grad()
                d_loss.backward()
                self.discriminator_optimizer.step()

            d_prec1 = accuracy(a_prob.data, self.sensitive_label.data)
            d_prec5 = accuracy(a_prob.data, self.sensitive_label.data,
            topk=(int(np.min([5, self.n_sensitive_class])),))
            self.d_losses.update(d_loss.data.item(), batch_size)
            self.d_top1.update(d_prec1[0], batch_size)
            self.d_top5.update(d_prec5[0], batch_size)

            if iteration % print_interval == 0:
                print(string + '_Epoch:[{0}][{1}/{2}]_|'
                        '_T_Loss:_{3:.2f}_|'
                        '_E_Loss:_{4:.2f}_|'
                        '_Loss:_{5:.2f}_|'
                        '_T_Prec:_{6:.2f}_|'
                        '_T_Prec5:_{7:.2f}_|'
                        '_D_Loss:_{8:.2f}_|'
                        '_D_Prec:_{9:.2f}_|'
                        '_D_Prec5:_{10:.2f}_|'
                        '_D_Entropy:_{11:.2f}_|'
                    .format(
                    epoch, batch_idx, data_size,
                    float(self.t_losses.avg), float(self.e_losses.avg),
                    float(self.losses.avg),float(self.t_top1.avg.item()),
                    float(self.t_top5.avg.item()), float(self.d_losses.avg),
                    float(self.d_top1.avg.item()), float(self.d_top5.avg.item()),
                    float(self.entropy_losses.avg)))

        else:
```

```python
            if iteration % print_interval == 0:
                print(string + '_Epoch:[{0}][{1}/{2}]_|'
                      '_T_Loss:_{3:.2f}_|'
                      '_T_Prec:_{4:.2f}_|'
                      '_T_Prec5:_{5:.2f}_|'
                      .format(
                      epoch, batch_idx, data_size,
                      float(self.t_losses.avg), float(self.t_top1.avg.item()),
                      float(self.t_top5.avg.item())))

    return self.losses.avg, self.t_top1.avg, self.t_top5.avg, self.d_losses.avg,
    self.d_top1.avg, self.d_top5.avg, self.entropy_losses.avg

def train(self):
    self.logger = Logger(os.path.join('checkpoint/',
    self.log_file_name), title='Problem')
    self.logger.set_names(['LR', 'Train-Loss', 'Test-Loss',
    'Train-Acc.', 'Train-Acc5.', 'Test_Acc.', 'Test_Acc5.',
                          'D-Train_Loss', 'D-Test_Loss',
                          'D-Train_Acc.', 'D-Train_Acc5.',
                          'D-Test_Acc.', 'D-Test_Acc5.',
                          'D-Train-Entropy',
                          'D-Test-Entropy'])

    for epoch in range(self.start_epoch, self.total_epoch):
        print('\nEpoch:_%d' % epoch)

        train_loss, train_acc, train_acc5, d_train_loss, d_train_acc,
        d_train_acc5, d_train_entropy =
        self.perform_epoch(epoch=epoch, test_flag=False)

        with torch.no_grad():
            test_loss, test_acc, test_acc5, d_test_loss, d_test_acc,
            d_test_acc5, d_test_entropy = self.perform_epoch(epoch=epoch,
                test_flag=True)

        self.logger.append([self.optimizer.param_groups[0]['lr'], float(train_los
                    float(test_loss),
                       float(train_acc), float(train_acc5),
                        float(test_acc), float(test_acc5),
                        float(d_train_loss),
                        float(d_test_loss),
                        float(d_train_acc),
                        float(d_train_acc5),
                        float(d_test_acc),
                        float(d_test_acc5),
                        float(d_train_entropy),
                        float(d_test_entropy)])

        # it is optimum only when we reach the end of the game by optimization,
        # any other value e.g., current discriminator feedback is non-optimal
        if (epoch + 1) % 10:
            print('Saving..')   # Save checkpoint.
            state = {
                'net': self.net.module if self.use_cuda else self.net,
                'state_dict': self.net.state_dict(),
                'acc': test_acc,
                'epoch': epoch,
```

```python
                    'optimizer': self.optimizer.state_dict()
                }
                if not os.path.isdir('checkpoint'):
                    os.mkdir('checkpoint')
                torch.save(state, 'checkpoint/' +
                self.checkpoint_filename + '.ckpt')
                self.best_acc = test_acc
                self.best_loss = test_loss

        self.logger.close()
        print("Done")

    def perform_epoch_adversary(self, epoch, test_flag=False):
        if test_flag:
            self.adv_net.eval()
            loader = self.test_loader
            str = "Test"
            print_interval = self.print_interval_test
        else:
            self.adv_net.train()
            loader = self.train_loader
            str = "Train"
            print_interval = self.print_interval_train

        self.net.eval()
        iteration = 0
        self.adv_losses.reset()
        self.adv_top1.reset()
        self.adv_top5.reset()
        self.entropy_losses.reset()

        for batch_idx, (inputs, target_label, sensitive_label) in enumerate(loader):
            batch_size = inputs.size(0)
            iteration += 1
            if self.data.name == 'mnist':
                inputs = torch.unsqueeze(inputs, 1).float()

            self.inputs.resize_(inputs.size()).copy_(inputs)
            self.target_label.resize_(target_label.size()).copy_(target_label)
            self.sensitive_label.resize_(
            sensitive_label.size()).copy_(sensitive_label)

            with torch.no_grad():
                outputs, z, _ = self.net(self.inputs)

            d_outputs, _, prob = self.adv_net(z.detach())
            d_loss = self.cross_entropy_loss(d_outputs, self.sensitive_label)

            with torch.no_grad():
                entropy_loss = -self.entropy_loss(prob)

            if not test_flag:
                self.adv_optimizer.zero_grad()
                d_loss.backward()
                self.adv_optimizer.step()

            d_prec1 = accuracy(prob.data, self.sensitive_label.data)
            d_prec5 = accuracy(prob.data, self.sensitive_label.data,
            topk=(int(np.min([5, self.n_sensitive_class])),))
```

```python
                self.adv_losses.update(d_loss.data.item(), batch_size)
                self.adv_top1.update(d_prec1[0], batch_size)
                self.adv_top5.update(d_prec5[0], batch_size)
                self.entropy_losses.update(entropy_loss.data.item(), batch_size)

                if iteration % print_interval == 0:
                    print(str + ' Epoch:[{0}][{1}/{2}] |'
                          ' T_Loss: {3:.5f} |'
                          ' T_Prec: {4:.2f} |'
                          ' T5_Prec: {5:.2f} |'
                          ' Entropy: {6:.3f} |'
                          .format(
                          epoch, batch_idx, len(self.train_loader),
                          float(self.adv_losses.avg),
                          float(self.adv_top1.avg.item()),
                          float(self.adv_top5.avg),
                          float(self.entropy_losses.avg)))

        return self.adv_losses.avg, self.adv_top1.avg,
        self.adv_top5.avg, self.entropy_losses.avg

    def train_adversary(self, model_filename=None, total_epoch=100):
        self.adv_logger = Logger(os.path.join('checkpoint/',
        self.adv_log_file_name), title='Problem')
        self.adv_logger.set_names(['LR', 'Train-Loss', 'Test-Loss', 'Train Acc.',
        'Train Acc5.', 'Test Acc.', 'Test Acc5.',
                                  'Train Entropy','Test Entropy'])

        self.adv_best_acc = 0
        scheduler = CosineAnnealingLR(self.adv_optimizer,
        T_max=total_epoch, eta_min=1e-7)
        if model_filename is not None:
            checkpoint = torch.load(os.path.join('checkpoint/', model_filename))
            self.net = checkpoint['net']
            self.net.eval()

        for epoch in range(total_epoch):
            print('\nEpoch: %d' % epoch)
            scheduler.step()
            train_loss, train_acc, train_acc5, train_entropy =
                self.perform_epoch_adversary(epoch=epoch, test_flag=False)
            with torch.no_grad():
                test_loss, test_acc, test_acc5, test_entropy =
                self.perform_epoch_adversary(epoch=epoch, test_flag=True)

            self.adv_logger.append([self.adv_optimizer.param_groups[0]['lr'],
            float(train_loss), float(test_loss), float(train_acc),
                        float(train_acc5), float(test_acc),
                        float(test_acc5),
                        float(train_entropy),
                        float(test_entropy)])
            # Save checkpoint.
            if test_acc > self.adv_best_acc:
                print('Saving..')
                state = {
                    'net': self.adv_net.module if self.use_cuda else self.adv_net,
                    'state_dict': self.adv_net.state_dict(),
                    'acc': test_acc,
                    'epoch': epoch,
```

```python
                'optimizer': self.adv_optimizer.state_dict()
            }
            if not os.path.isdir('checkpoint'):
                os.mkdir('checkpoint')
            torch.save(state, 'checkpoint/' + self.adv_checkpoint_filename)
            self.adv_best_acc = test_acc

    self.adv_logger.close()
    print("Adversary Done.")

def perform_epoch_target(self, epoch, test_flag=False):
    if test_flag:
        self.target_net.eval()
        loader = self.test_loader
        str = "Test"
        print_interval = self.print_interval_test
    else:
        self.target_net.train()
        loader = self.train_loader
        str = "Train"
        print_interval = self.print_interval_train

    self.net.eval()
    iteration = 0
    self.target_losses.reset()
    self.target_top1.reset()
    self.target_top5.reset()
    self.target_entropy_losses.reset()

    for batch_idx, (inputs, target_label, sensitive_label) in enumerate(loader):
        batch_size = inputs.size(0)
        iteration += 1
        if self.data.name == 'mnist':
            inputs = torch.unsqueeze(inputs, 1).float()

        self.inputs.resize_(inputs.size()).copy_(inputs)
        self.target_label.resize_(target_label.size()).copy_(target_label)
        self.sensitive_label.resize_(
        sensitive_label.size()).copy_(sensitive_label)

        with torch.no_grad():
            outputs, z, _ = self.net(self.inputs)

        d_outputs, _, prob = self.target_net(z.detach())
        d_loss = self.cross_entropy_loss(d_outputs, self.target_label)

        with torch.no_grad():
            entropy_loss = -self.entropy_loss(prob)

        if not test_flag:
            self.target_optimizer.zero_grad()
            d_loss.backward()
            self.target_optimizer.step()

        d_prec1 = accuracy(prob.data, self.target_label.data)
        d_prec5 = accuracy(prob.data, self.target_label.data,
        topk=(int(np.min([5, self.n_target_class])),))
        self.target_losses.update(d_loss.data.item(), batch_size)
        self.target_top1.update(d_prec1[0], batch_size)
```

```python
            self.target_top5.update(d_prec5[0], batch_size)
            self.target_entropy_losses.update(entropy_loss.data.item(), batch_size)

            if iteration % print_interval == 0:
                print(str + ' Epoch:[{0}][{1}/{2}] |'
                      ' T_Loss: {3:.5f} |'
                      ' T_Prec: {4:.2} |'
                      ' T5_Prec: {5:.2f} |'
                      ' Entropy: {6:.3f} |'
                    .format(
                    epoch, batch_idx, len(loader),
                    float(self.target_losses.avg),
                    float(self.target_top1.avg.item()),
                    float(self.target_top5.avg),
                    float(self.target_entropy_losses.avg)))

        return self.target_losses.avg, self.target_top1.avg,
        self.target_top5.avg, self.target_entropy_losses.avg

    def train_target(self, model_filename=None, total_epoch=100):
        self.target_logger = Logger(os.path.join('checkpoint/',
        self.target_log_file_name), title='Problem')
        self.target_logger.set_names(['LR', 'Train-Loss',
        'Test-Loss', 'Train Acc.', 'Train Acc5.', 'Test Acc.', 'Test Acc5.',
                                      'Train Entropy','Test Entropy'])

        self.target_best_acc = 0
        scheduler = CosineAnnealingLR(self.target_optimizer,
        T_max=total_epoch, eta_min=1e-7)
        if model_filename is not None:
            assert os.path.isdir('checkpoint'), 'Error: no
            checkpoint directory found!'
            checkpoint = torch.load(os.path.join('checkpoint/', model_filename))
            self.net = checkpoint['net']
            self.net.eval()

        for epoch in range(total_epoch):
            print('\nEpoch: %d' % epoch)
            scheduler.step()
            train_loss, train_acc, train_acc5, train_entropy =
            self.perform_epoch_target(epoch=epoch, test_flag=False)
            with torch.no_grad():
                test_loss, test_acc, test_acc5, test_entropy =
                self.perform_epoch_target(epoch=epoch, test_flag=True)

            self.target_logger.append([self.target_optimizer.param_groups[0]['lr'],

                            float(train_acc5), float(test_acc),
                            float(test_acc5),
                            float(train_entropy),
                            float(test_entropy)])

            if test_acc > self.target_best_acc:
                print('Saving..')  # Save checkpoint.
                state = {
                    'net': self.target_net.module
                    if self.use_cuda else self.target_net,
                    'state_dict': self.target_net.state_dict(),
                    'acc': test_acc,
```

Table 8: Datasets used in our DG experiments, with sample images for each of them. This table is taken from Gulrajani and Lopez-Paz (2020).

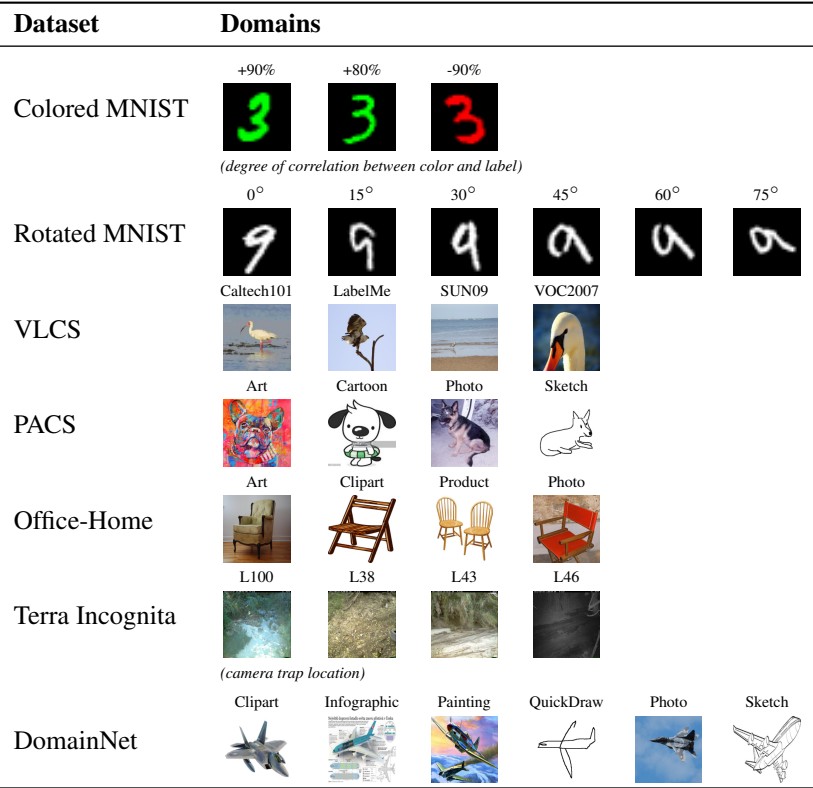

```
            'epoch': epoch,
            'optimizer': self.target_optimizer.state_dict()
        }
        if not os.path.isdir('checkpoint'):
            os.mkdir('checkpoint')
        torch.save(state, 'checkpoint/' + self.target_checkpoint_filename)
        self.target_best_acc = test_acc

    self.target_logger.close()
    print("Target_Done")
```

Table 7: Main characteristics of the datasets used in our fair representation learning experiments.

| Dataset | Support of $D$ | Target Variable | Dataset Size | Input Size | Majority Sensitive | Majority Target |
|---------|----------------|------------------|--------------|------------|--------------------|------------------|
| Adult | { male, female } | Income > 50,000$ | 45,222 | 14 | 67% | 75% |
| German | { male, female } | Good or bad credit | 1,000 | 20 | 69% | 71% |

## E    DG SUPPLEMENTS

**Compute Resources**    We run the 10,560 jobs on NVIDIA GEFORCE RTX 2080 TI GPUs as well as NVIDIA TITAN RTX GPUs for the more resource intensive jobs.

**Baseline models**    We compare our algorithms to the following existing algorithms:

Table 9: Description of the datasets used in our DG experiments

| Dataset Name | Support of $D$ | Number of Samples | Image Dimensions | Number of Classes |
|---|---|---|---|---|
| ColoredMNIST (Arjovsky et al., 2019) | $\{0.1, 0.3, 0.9\}$ | $70,000$ | $(2, 28, 28)$ | 2 |
| RotatedMNIST (Ghifary et al., 2015) | $\{0, 15, 30, 45, 60, 75\}$ | $70,000$ | $(1, 28, 28)$ | 10 |
| VLCS (Fang et al., 2013) | $\{$Caltech101, LabelMe, SUN09, VOC2007$\}$ | $10,729$ | $(3, 224, 224)$ | 5 |
| PACS (Li et al., 2017) | $\{$art, cartoons, photos, sketches$\}$ | $9,991$ | $(3, 224, 224)$ | 7 |
| OfficeHome (Venkateswara et al., 2017) | $\{$art, clipart, product, real$\}$ | $15,588$ | $(3, 224, 224)$ | 65 |
| TerraIncognita (Beery et al., 2018) | $\{$L100, L38, L43, L46$\}$ | $24,788$ | $(3, 224, 224)$ | 10 |
| DomainNet (Peng et al., 2019) | $\{$clipart, infograph, painting, quickdraw, real, sketch$\}$ | $586,575$ | $(3, 224, 224)$ | 345 |

- Empirical Risk Minimization (ERM, Vapnik (1998)), where the sum of errors is minimized across domains.

- Group Distributionally Robust Optimization (DRO, Sagawa et al. (2019)), where low performing domains are giving an increasing weight during training.

- Inter-domain Mixup (Mixup, Yan et al. (2020)).

- Meta-Learning for Domain Generalization (MLDG, Li et al. (2018a)).

- Algorithms based on matching the latent distribution across domains:

    - Domain-Adversarial Neural Networks (DANN, Ganin et al. (2016)), where the distributional distance is an adversarial network.

    - Class-conditional DANN (C-DANN, Li et al. (2018d)), which is a variant of DANN matching the class conditional distributions across domains.

    - CORAL Sun and Saenko (2016), which aligns the mean and covariance of latent distributions.

    - MMD Li et al. (2018b), which uses the MMD distance.

- Invariant Risk Minimization (IRM Arjovsky et al. (2019)), which looks for a representation whose optimal linear classifier on top of the representation matches across domains.

- Style Agnostic Networks (SagNet, Nam et al. (2021)), which tries to reduce style bias of CNNs.

- Adaptive Risk Minimization (ARM, Zhang et al. (2020)), which is based on meta-learning.

- Variance Risk Extrapolation (VREx, Krueger et al. (2021)), where they enforce the training risk to be similar across domains.

- Representation Self-Challenging (RSC, Huang et al. (2020)).

**Implementation**  To be more concrete, we change the code that computes the distributional distance penalty from this:

```
for i in range(nmb):
    for j in range(i + 1, nmb):
        penalty += self.dist_loss(features[i], features[j])

if nmb > 1:
    penalty /= (nmb * (nmb - 1) / 2)
```

to this:

```
first = None
second = None

for i in range(nmb):
    slice = random.randint(0, len(features[i]))

    if first is None:                                           .
        first = features[i][:slice]
        second = features[i][slice:]
    else:
        first = torch.cat((first, features[i][:slice]), 0)
        second = torch.cat((second, features[i][slice:]), 0)

penalty = self.dist_loss(first, second)
```

Here is the concrete full class of our CausIRL with MMD model:

```
class CausIRL_MMD(ERM):
    def __init__(self, input_shape, num_classes, num_domains, hparams):
        super(CausIRL_MMD, self).__init__(input_shape, num_classes, num_domains,
                                 hparams)
        self.kernel_type = "gaussian"

    def my_cdist(self, x1, x2):
        x1_norm = x1.pow(2).sum(dim=-1, keepdim=True)
        x2_norm = x2.pow(2).sum(dim=-1, keepdim=True)
        res = torch.addmm(x2_norm.transpose(-2, -1),
                        x1,
                        x2.transpose(-2, -1), alpha=-2).add_(x1_norm)
        return res.clamp_min_(1e-30)

    def gaussian_kernel(self, x, y, gamma=[0.001, 0.01, 0.1, 1, 10, 100,
                                            1000]):
        D = self.my_cdist(x, y)
        K = torch.zeros_like(D)

        for g in gamma:
            K.add_(torch.exp(D.mul(-g)))

        return K

    def mmd(self, x, y):
        Kxx = self.gaussian_kernel(x, x).mean()
        Kyy = self.gaussian_kernel(y, y).mean()
        Kxy = self.gaussian_kernel(x, y).mean()
        return Kxx + Kyy - 2 * Kxy

    def update(self, minibatches, unlabeled=None):
        objective = 0
        penalty = 0
        nmb = len(minibatches)

        features = [self.featurizer(xi) for xi, _ in minibatches]
        classifs = [self.classifier(fi) for fi in features]
        targets = [yi for _, yi in minibatches]

        first = None
        second = None
```

Table 10: DG experimental results for the training-domain validation selection method.

| Algorithm | ColoredMNIST | RotatedMNIST | VLCS | PACS | OfficeHome | TerraIncognita | DomainNet | Avg |
|---|---|---|---|---|---|---|---|---|
| CausIRL with CORAL (ours) | $51.7 \pm 0.1$ | $97.9 \pm 0.1$ | $77.5 \pm 0.6$ | $85.8 \pm 0.1$ | $68.6 \pm 0.3$ | $47.3 \pm 0.8$ | $41.9 \pm 0.1$ | 67.3 |
| CORAL | $51.5 \pm 0.1$ | $98.0 \pm 0.1$ | $78.8 \pm 0.6$ | $86.2 \pm 0.3$ | $68.7 \pm 0.3$ | $47.6 \pm 1.0$ | $41.5 \pm 0.1$ | 67.5 |
| CausIRL with MMD (ours) | $51.6 \pm 0.1$ | $97.9 \pm 0.0$ | $77.6 \pm 0.4$ | $84.0 \pm 0.8$ | $65.7 \pm 0.6$ | $46.3 \pm 0.9$ | $40.3 \pm 0.2$ | 66.2 |
| MMD | $51.5 \pm 0.2$ | $97.9 \pm 0.0$ | $77.5 \pm 0.9$ | $84.6 \pm 0.5$ | $66.3 \pm 0.1$ | $42.2 \pm 1.6$ | $23.4 \pm 9.5$ | 63.3 |
| ERM | $51.5 \pm 0.1$ | $98.0 \pm 0.0$ | $77.5 \pm 0.4$ | $85.5 \pm 0.2$ | $66.5 \pm 0.3$ | $46.1 \pm 1.8$ | $40.9 \pm 0.1$ | 66.6 |
| IRM | $52.0 \pm 0.1$ | $97.7 \pm 0.1$ | $78.5 \pm 0.5$ | $83.5 \pm 0.8$ | $64.3 \pm 2.2$ | $47.6 \pm 0.8$ | $33.9 \pm 2.8$ | 65.4 |
| GroupDRO | $52.1 \pm 0.0$ | $98.0 \pm 0.0$ | $76.7 \pm 0.6$ | $84.4 \pm 0.8$ | $66.0 \pm 0.7$ | $43.2 \pm 1.1$ | $33.3 \pm 0.2$ | 64.8 |
| Mixup | $52.1 \pm 0.2$ | $98.0 \pm 0.1$ | $77.4 \pm 0.6$ | $84.6 \pm 0.6$ | $68.1 \pm 0.3$ | $47.9 \pm 0.8$ | $39.2 \pm 0.1$ | 66.7 |
| MLDG | $51.5 \pm 0.1$ | $97.9 \pm 0.0$ | $77.2 \pm 0.4$ | $84.9 \pm 1.0$ | $66.8 \pm 0.6$ | $47.7 \pm 0.9$ | $41.2 \pm 0.1$ | 66.7 |
| DANN | $51.5 \pm 0.3$ | $97.8 \pm 0.1$ | $78.6 \pm 0.4$ | $83.6 \pm 0.4$ | $65.9 \pm 0.6$ | $46.7 \pm 0.5$ | $38.3 \pm 0.1$ | 66.1 |
| CDANN | $51.7 \pm 0.1$ | $97.9 \pm 0.1$ | $77.5 \pm 0.1$ | $82.6 \pm 0.9$ | $65.8 \pm 1.3$ | $45.8 \pm 1.6$ | $38.3 \pm 0.3$ | 65.6 |
| MTL | $51.4 \pm 0.1$ | $97.9 \pm 0.0$ | $77.2 \pm 0.4$ | $84.6 \pm 0.5$ | $66.4 \pm 0.5$ | $45.6 \pm 1.2$ | $40.6 \pm 0.1$ | 66.2 |
| SagNet | $51.7 \pm 0.0$ | $98.0 \pm 0.0$ | $77.8 \pm 0.5$ | $86.3 \pm 0.2$ | $68.1 \pm 0.1$ | $48.6 \pm 1.0$ | $40.3 \pm 0.1$ | 67.2 |
| ARM | $56.2 \pm 0.2$ | $98.2 \pm 0.1$ | $77.6 \pm 0.3$ | $85.1 \pm 0.4$ | $64.8 \pm 0.3$ | $45.5 \pm 0.3$ | $35.5 \pm 0.2$ | 66.1 |
| VREx | $51.8 \pm 0.1$ | $97.9 \pm 0.1$ | $78.3 \pm 0.2$ | $84.9 \pm 0.6$ | $66.4 \pm 0.6$ | $46.4 \pm 0.6$ | $33.6 \pm 2.9$ | 65.6 |
| RSC | $51.7 \pm 0.2$ | $97.6 \pm 0.1$ | $77.1 \pm 0.5$ | $85.2 \pm 0.9$ | $65.5 \pm 0.9$ | $46.6 \pm 1.0$ | $38.9 \pm 0.5$ | 66.1 |

```python
for i in range(nmb):
    objective += F.cross_entropy(classifs[i] + 1e-16, targets[i])
    slice = random.randint(0, len(features[i]))
    if first is None:
        first = features[i][:slice]
        second = features[i][slice:]
    else:
        first = torch.cat((first, features[i][:slice]), 0)
        second = torch.cat((second, features[i][slice:]), 0)
if len(first) > 1 and len(second) > 1:
    penalty = torch.nan_to_num(self.mmd(first, second))
else:
    penalty = torch.tensor(0)

objective /= nmb

self.optimizer.zero_grad()
(objective + (self.hparams['mmd_gamma']*penalty)).backward()
self.optimizer.step()

if torch.is_tensor(penalty):
    penalty = penalty.item()

return {'loss': objective.item(), 'penalty': penalty}
```

### E.0.1 MODEL SELECTION: TRAINING-DOMAIN VALIDATION SET

We present here the results of our DG experiments for the training-domain validation model selection method. Result are summarized in Table 10. For CausIRL with CORAL, the overall performance is slightly below vanilla CORAL. CausIRL with CORAL especially underperforms CORAL on the PACS dataset. On the other hand, CausIRL with CORAL performs better than CORAL on DomainNet. For CausIRL with MMD, the overall performance is significantly better than MMD. This overperformance is mainly driven by the results on TerraIncognita and DomainNet, where for the latter we observe a leap in accuracy from $23.4\%$ to $40.3\%$.

### E.1 MODEL SELECTION: LEAVE-ONE-DOMAIN-OUT CROSS-VALIDATION

We here present the complete results for the leave-one-domain-out cross-validation model selection method in Table 11.

Table 11: DG experimental results for the leave-one-domain-out cross-validation model selection method.

| Algorithm | ColoredMNIST | RotatedMNIST | VLCS | PACS | OfficeHome | TerraIncognita | DomainNet | Avg |
|---|---|---|---|---|---|---|---|---|
| **CausIRL with CORAL (ours)** | $39.1 \pm 2.0$ | $97.8 \pm 0.1$ | $76.5 \pm 1.0$ | $83.6 \pm 1.2$ | $68.1 \pm 0.3$ | $47.4 \pm 0.5$ | $41.8 \pm 0.1$ | 64.9 |
| CORAL | $39.7 \pm 2.8$ | $97.8 \pm 0.1$ | $78.7 \pm 0.4$ | $82.6 \pm 0.5$ | $68.5 \pm 0.2$ | $46.3 \pm 1.7$ | $41.1 \pm 0.1$ | 65.0 |
| **CausIRL with MMD (ours)** | $36.9 \pm 0.2$ | $97.6 \pm 0.1$ | $78.2 \pm 0.9$ | $84.0 \pm 0.9$ | $65.1 \pm 0.7$ | $47.9 \pm 0.3$ | $38.9 \pm 0.8$ | 64.1 |
| MMD | $36.8 \pm 0.1$ | $97.8 \pm 0.1$ | $77.3 \pm 0.5$ | $83.2 \pm 0.2$ | $60.2 \pm 5.2$ | $46.5 \pm 1.5$ | $23.4 \pm 9.5$ | 60.7 |
| ERM | $36.7 \pm 0.1$ | $97.7 \pm 0.0$ | $77.2 \pm 0.4$ | $83.0 \pm 0.7$ | $65.7 \pm 0.5$ | $41.4 \pm 1.4$ | $40.6 \pm 0.2$ | 63.2 |
| IRM | $40.3 \pm 4.2$ | $97.0 \pm 0.2$ | $76.3 \pm 0.6$ | $81.5 \pm 0.8$ | $64.3 \pm 1.5$ | $41.2 \pm 3.6$ | $33.5 \pm 3.0$ | 62.0 |
| GroupDRO | $36.8 \pm 0.1$ | $97.6 \pm 0.1$ | $77.9 \pm 0.5$ | $83.5 \pm 0.2$ | $65.2 \pm 0.2$ | $44.9 \pm 1.4$ | $33.0 \pm 0.3$ | 62.7 |
| Mixup | $33.4 \pm 4.7$ | $97.8 \pm 0.0$ | $77.7 \pm 0.6$ | $83.2 \pm 0.4$ | $67.0 \pm 0.2$ | $48.7 \pm 0.4$ | $38.5 \pm 0.3$ | 63.8 |
| MLDG | $36.7 \pm 0.2$ | $97.6 \pm 0.0$ | $77.2 \pm 0.9$ | $82.9 \pm 1.7$ | $66.1 \pm 0.5$ | $46.2 \pm 0.9$ | $41.0 \pm 0.2$ | 64.0 |
| DANN | $40.7 \pm 2.3$ | $97.6 \pm 0.2$ | $76.9 \pm 0.4$ | $81.0 \pm 1.1$ | $64.9 \pm 1.2$ | $44.4 \pm 1.1$ | $38.2 \pm 0.2$ | 63.4 |
| CDANN | $39.1 \pm 4.4$ | $97.5 \pm 0.2$ | $77.5 \pm 0.2$ | $78.8 \pm 2.2$ | $64.3 \pm 1.7$ | $39.9 \pm 3.2$ | $38.0 \pm 0.1$ | 62.2 |
| MTL | $35.0 \pm 1.7$ | $97.8 \pm 0.1$ | $76.6 \pm 0.5$ | $83.7 \pm 0.4$ | $65.7 \pm 0.5$ | $44.9 \pm 1.2$ | $40.6 \pm 0.1$ | 63.5 |
| SagNet | $36.5 \pm 0.1$ | $94.0 \pm 3.0$ | $77.5 \pm 0.3$ | $82.3 \pm 0.1$ | $67.6 \pm 0.3$ | $47.2 \pm 0.9$ | $40.2 \pm 0.2$ | 63.6 |
| ARM | $36.8 \pm 0.0$ | $98.1 \pm 0.1$ | $76.6 \pm 0.5$ | $81.7 \pm 0.2$ | $64.4 \pm 0.2$ | $42.6 \pm 2.7$ | $35.2 \pm 0.1$ | 62.2 |
| VREx | $36.9 \pm 0.3$ | $93.6 \pm 3.4$ | $76.7 \pm 1.0$ | $81.3 \pm 0.9$ | $64.9 \pm 1.3$ | $37.3 \pm 3.0$ | $33.4 \pm 3.1$ | 60.6 |
| RSC | $36.5 \pm 0.2$ | $97.6 \pm 0.1$ | $77.5 \pm 0.5$ | $82.6 \pm 0.7$ | $65.8 \pm 0.7$ | $40.0 \pm 0.8$ | $38.9 \pm 0.5$ | 62.7 |

Table 12: DG experimental results for the test-domain validation set model selection method.

| Algorithm | ColoredMNIST | RotatedMNIST | VLCS | PACS | OfficeHome | TerraIncognita | DomainNet | Avg |
|---|---|---|---|---|---|---|---|---|
| **CausIRL with CORAL (ours)** | $58.4 \pm 0.3$ | $98.0 \pm 0.1$ | $78.2 \pm 0.1$ | $87.6 \pm 0.1$ | $67.7 \pm 0.2$ | $53.4 \pm 0.4$ | $42.1 \pm 0.1$ | 69.4 |
| CORAL | $58.6 \pm 0.5$ | $98.0 \pm 0.0$ | $77.7 \pm 0.2$ | $87.1 \pm 0.5$ | $68.4 \pm 0.2$ | $52.8 \pm 0.2$ | $41.8 \pm 0.1$ | 69.2 |
| **CausIRL with MMD (ours)** | $63.7 \pm 0.8$ | $97.9 \pm 0.1$ | $78.1 \pm 0.1$ | $86.6 \pm 0.7$ | $65.2 \pm 0.6$ | $52.2 \pm 0.3$ | $40.6 \pm 0.2$ | 69.2 |
| MMD | $63.3 \pm 1.3$ | $98.0 \pm 0.1$ | $77.9 \pm 0.1$ | $87.2 \pm 0.1$ | $66.2 \pm 0.3$ | $52.0 \pm 0.4$ | $23.5 \pm 9.4$ | 66.9 |
| ERM | $57.8 \pm 0.2$ | $97.8 \pm 0.1$ | $77.6 \pm 0.3$ | $86.7 \pm 0.3$ | $66.4 \pm 0.5$ | $53.0 \pm 0.3$ | $41.3 \pm 0.1$ | 68.7 |
| IRM | $67.7 \pm 1.2$ | $97.5 \pm 0.2$ | $76.9 \pm 0.6$ | $84.5 \pm 1.1$ | $63.0 \pm 2.7$ | $50.5 \pm 0.7$ | $28.0 \pm 5.1$ | 66.9 |
| GroupDRO | $61.1 \pm 0.9$ | $97.9 \pm 0.1$ | $77.4 \pm 0.5$ | $87.1 \pm 0.1$ | $66.2 \pm 0.6$ | $52.4 \pm 0.1$ | $33.4 \pm 0.3$ | 67.9 |
| Mixup | $58.4 \pm 0.2$ | $98.0 \pm 0.1$ | $78.1 \pm 0.3$ | $86.8 \pm 0.3$ | $68.0 \pm 0.2$ | $54.4 \pm 0.3$ | $39.6 \pm 0.1$ | 69.0 |
| MLDG | $58.2 \pm 0.4$ | $97.8 \pm 0.1$ | $77.5 \pm 0.1$ | $86.8 \pm 0.4$ | $66.6 \pm 0.3$ | $52.0 \pm 0.1$ | $41.6 \pm 0.1$ | 68.7 |
| DANN | $57.0 \pm 1.0$ | $97.9 \pm 0.1$ | $79.7 \pm 0.5$ | $85.2 \pm 0.2$ | $65.3 \pm 0.8$ | $50.6 \pm 0.4$ | $38.3 \pm 0.1$ | 67.7 |
| CDANN | $59.5 \pm 2.0$ | $97.9 \pm 0.0$ | $79.9 \pm 0.2$ | $85.8 \pm 0.8$ | $65.3 \pm 0.5$ | $50.8 \pm 0.6$ | $38.5 \pm 0.2$ | 68.2 |
| MTL | $57.6 \pm 0.3$ | $97.9 \pm 0.1$ | $77.7 \pm 0.5$ | $86.7 \pm 0.2$ | $66.5 \pm 0.4$ | $52.2 \pm 0.4$ | $40.8 \pm 0.1$ | 68.5 |
| SagNet | $58.2 \pm 0.3$ | $97.9 \pm 0.0$ | $77.6 \pm 0.1$ | $86.4 \pm 0.4$ | $67.5 \pm 0.2$ | $52.5 \pm 0.4$ | $40.8 \pm 0.2$ | 68.7 |
| ARM | $63.2 \pm 0.7$ | $98.1 \pm 0.1$ | $77.8 \pm 0.3$ | $85.8 \pm 0.2$ | $64.8 \pm 0.4$ | $51.2 \pm 0.5$ | $36.0 \pm 0.2$ | 68.1 |
| VREx | $67.0 \pm 1.3$ | $97.9 \pm 0.1$ | $78.1 \pm 0.2$ | $87.2 \pm 0.6$ | $65.7 \pm 0.3$ | $51.4 \pm 0.5$ | $30.1 \pm 3.7$ | 68.2 |
| RSC | $58.5 \pm 0.5$ | $97.6 \pm 0.1$ | $77.8 \pm 0.6$ | $86.2 \pm 0.5$ | $66.5 \pm 0.6$ | $52.1 \pm 0.2$ | $38.9 \pm 0.6$ | 68.2 |

### E.1.1 MODEL SELECTION: TEST-DOMAIN VALIDATION SET (ORACLE)

Finally, we here look at the DG experiment results for the test-domain validation set model selection. The results are summarized in Table 12. This setting is less realistic as we have access to test samples during training, but it is still useful as it shows the best possible model for each algorithm. It allows us to evaluate whether there is headroom for improvement for each algorithm and to see which algorithm has the inductive bias that more closely fit the task.

For both CausIRL with CORAL and CausIRL with MMD, we observe a better overall performance compared to their vanilla counterparts. We even have that CausIRL with CORAL is the best overall performing algorithm among the evaluated algorithms. Once again, we observe a large difference in performance on DomainNet between MMD and CausIRL with MMD, going from an average accuracy of 23.5% to 40.6%. We also again have that CausIRL with CORAL is the best algorithm for DomainNet compared to all the other algorithms, which explains why we perform well on this dataset.

