# OpenReview forum: "Invariant Causal Mechanisms through Distribution Matching"
_ICLR.cc/2022/Conference — ICLR 2022 Submitted_

### Official Review · Reviewer_H74w · 2021-11-02

**Correctness:** 3
**Technical Novelty And Significance:** 2
**Empirical Novelty And Significance:** 2
**Recommendation:** 3
**Confidence:** 3

**Main Review:**

The paper is well-written and easy to read. The proposed algorithm is pretty straightforward.
I am however not sure if the proposed algorithm would be successful in learning invariant representations. As a toy example, imagine two datasets from different domains with only two features: x1 and x2. Now let’s say the distribution of x1 is the same for both datasets but not for x2. In other words, x2 is the style variable that should be eliminated from the learned representation. Now if we sample from these two datasets and create two mixed datasets (following the proposed algorithm) we would have two datasets with almost identical distributions (likely multi-modal). I can’t see how minimizing the distance between the distributions of these two mixed datasets would eliminate the style variable.


**Summary Of The Paper:**

This work provides a causal perspective and new algorithm for learning invariant representations from multiple domain datasets. This work introduces the notion of style variables and shows theoretically that being invariant to the domain index actually leads to invariance to the style variables. The extensive experiments demonstrate the usefulness of the proposed algorithm in various tasks.


**Summary Of The Review:**

The toy example described above shows that the proposed algorithm should not be able to learn invariant representations. There is however the possibility that I have misunderstood parts of the paper. I’m willing to increase my score should the authors address the above-mentioned issue.

---

> ### Author Response · Authors · 2021-11-16
> **Response to Review of Paper3171 by Reviewer H74w**
>
> We thank the reviewer for carefully assessing the soundness of our proposed algorithm.
>
> Unfortunately, it seems that the presentation of the algorithm and our theoretical are not clear enough. We do not claim that our algorithm will provably learn a representation that is invariant to the style variables. Depending on the distribution of the variable x2 in your counter-example, we may indeed not end up with a representation invariant to x2. However, what we show is that invariance to the domain index is necessary, and may be sufficient in some restricted settings. We nevertheless conjecture that our proposed algorithm may lead to stronger invariance to the style variables. To do so, we enforce the representation to be invariant to soft interventions on the distribution of the domain index, a.k.a, at each step of optimization, we create two new datasets with different mixtures of the domains.
>
> Please also note that your proposed toy example is very close to the toy example we test in the Appendix, where we show that our algorithm actually learns invariance to the domain index in this toy example.

---

### Official Review · Reviewer_PYVV · 2021-11-03

**Correctness:** 3
**Technical Novelty And Significance:** 2
**Empirical Novelty And Significance:** 3
**Recommendation:** 5
**Confidence:** 4

**Main Review:**

For the causal analysis part, authors propose a causal graph, which is a slight modification of the causal generative process proposed by Suter et al.(2019). Authors claimed “our proposal naturally captures most of the existing invariant representation learning tasks and datasets”, but I did not find much content showing how this causal graph could do so. Recent works have proposed different causal graphs for similar tasks (DG, fairness, etc.) and each has certain reasoning about the causal graph. Note that causal graph is untestable from data, so I keep neutral on this part.

The rest of the analysis, e.g., theorems, is based on the proposed causal graph, which is somewhat straightforward. Theorem 4.2 is misleading: a number of domains are firstly given but the statements do not specify what the are the domains. Indeed, after checking the proof, it really means all the possible domains or interventions. Then this is a trivial result according to the definition.

Conjecture 4.3 is not sufficiently motivating for the algorithm. Recall group DRO. The minimax problem is defined wrt the mixture of availability domains or distributions, but the optimum can be achieved by only considering the finite available domains, so there is no need to consider the mixture into optimization. Similarly, if the given domains satisfy the so called independence constraint, then any mixture does and vice versa. Thus, in theory, there is no need to consider the mixture processing in the algorithm. Please correct me I am wrong here.

Experiments show some improvement, but not very strong; see,e.g., the DG tasks. But that is OK from my side.



**Summary Of The Paper:**

This paper considers a causal framework for invariant representation learning, with some analysis about the invariance and independences based on the proposed causal graph. An algorithm is then proposed based on the above analysis and empirical result validates the performance of the proposed algorithm. The paper is well written.

**Summary Of The Review:**

Overall, the theoretical part (analysis, foundation for the algorithm) is weak. Given that the empirical results are not very strong, I cannot recommend an acceptance.

---

> ### Author Response · Authors · 2021-11-16
> **Response to Review of Paper3171 by Reviewer PYVV**
>
> We thank the reviewer for their balanced and insightful comments.
>
> We agree that the assumptions on the data generating graph are untestable and thus it is hard to evaluate which assumptions are realistic. Nevertheless, we considered that the graph proposed by Suter et al.(2019) was very sensical, and this is why we decided to slightly modify it for our setting instead of unnecessarily coming up with a new graph.
>
> Regarding the theory, it indeed straightly follows from our assumptions and definitions. Here the focus is mainly on the notion of style variables. We argue that to achieve better OOD generalization, existing models implicitly try to enforce invariance to the style variables through the proxy of the domain index. We show that in general, and without stronger assumptions, there is no guarantee that this goal will be achieved.
>
> Regarding conjecture 4.3 and the comments about mixture, the reviewer is right in stating that considering the mixtures or not is theoretically equivalent for invariance to the domain index. What we here want to argue is that enforcing invariance to the mixtures may lead to greater invariance to the style variables, which we assume should lead to better generalization. Moreover, as mixtures or not is theoretically equivalent for invariance to the domain index, there is no particular reason to prefer one over the other in practice. As our experiments show, enforcing invariance to the mixtures actually demonstrates surprisingly high performance.
>
> Lastly, we would like to emphasize the strength and consistency of our experimental results, especially on DomainBed. We compare our performance against 14 state-of-the-art baselines, and our two proposed models perform consistently well on all datasets and model selection methods (see Appendix also). We achieve state-of-the-art performance on multiple datasets as well as on average performance across datasets.

---

### Official Review · Reviewer_sMkT · 2021-11-12

**Correctness:** 3
**Technical Novelty And Significance:** 1
**Empirical Novelty And Significance:** 1
**Recommendation:** 1
**Confidence:** 5

**Main Review:**

- There are several fundamental misconceptions in this paper.
    -  Invariance is a property of causality that has been well established.
        - The invariance literature is in fact motivated and established through the causal graphs.
        - See [1] for a survey. For a textbook treatment, see chapter 2 of [2].
    -  Causal and Anti-causal
        - Factor models are different from causal models. Figure 1 is not a causal graph. It is a graphical model. A causal graph reflects the true data generating process in the world. "Background color" is not a variable in the world.
        - If I were to interpret figure 1 charitably, it is plausible that it actually means a human's perceiving process. However, that argument was not made in the paper.
        - The difference between how the world works and how humans perceive is the central debate between causal and anti-causal. It is not whether the task is image classification or not. The arguments such as the work like IRM is only about causal but not anti-causal are plainly wrong.
- The algorithm is reframing the IRM algorithm. The theorems are all tautological
    - Theorem 3.1 is just a conditional independent statement.
    - Theorem 4.1 and theorem 4.2 are circular statements. Invariance is defined (in def 3.2) with respect to total causal effect & conditional independence.
    - Conjecture 4.3 is factually correct. Those mixture environments are just environments/domains induced by soft intervention. It's also the setting of IRM. However, there is no argument for why that would increase OOD performance.
    - the algorithm in eq 1 is just reframing eq 1 in IRM.

- Writing
    - The authors did not introduce many concepts that are essential to their theoretical analysis. Examples include the do notation and the causal graph. The total causal effect is critical to the results, but it's only briefly introduced in the footnote.

- references
    - [1]
@article{scholkopf2021toward,
  title={Toward causal representation learning},
  author={Sch{\"o}lkopf, Bernhard and Locatello, Francesco and Bauer, Stefan and Ke, Nan Rosemary and Kalchbrenner, Nal and Goyal, Anirudh and Bengio, Yoshua},
  journal={Proceedings of the IEEE},
  volume={109},
  number={5},
  pages={612--634},
  year={2021},
  publisher={IEEE}
}
  -  [2]
@book{peters2017elements,
  title={Elements of causal inference: foundations and learning algorithms},
  author={Peters, Jonas and Janzing, Dominik and Sch{\"o}lkopf, Bernhard},
  year={2017},
  publisher={The MIT Press}
}

   -  IRM
@article{arjovsky2019invariant,
  title={Invariant risk minimization},
  author={Arjovsky, Martin and Bottou, L{\'e}on and Gulrajani, Ishaan and Lopez-Paz, David},
  journal={arXiv preprint arXiv:1907.02893},
  year={2019}
}

**Summary Of The Paper:**

- The paper claims to provide a unifying causal framework for invariance-based algorithms.
- It uses the graphical model in figure 1 to derive some conditions for achieving invariance.
- It also develops a new algorithm for invariance-based representation learning. The idea is based on conjecture 4.3, which hypothesizes that one can create new domains using mixtures of existing ones.


**Summary Of The Review:**

The paper has several fundamental issues, including a misunderstanding of the existing work.
Notably, the algorithm proposed in this paper is very similar to the algorithm in IRM.

---

> ### Author Response · Authors · 2021-11-19
> **Response to Review of Paper3171 by Reviewer sMkT**
>
> We thank reviewer sMkT for their feedback.
>
> We agree that factor models are different from causal models, however we disagree that figure 1 is not a causal model. It is even a precise causal model e.g. for the datasets of [1, 2]. Similarly to the motivation of our work, others likewise use the underlying causal graph of figure 1 as a motivation for their work e.g. [3]. For real-world applications the underlying causal graph is typically not known, yet we empirically demonstrate across multiple datasets that even the simplification and understanding derived from a simple causal graph as in figure 1 can significantly help.
>
> We agree that there are close connections between causality and invariance which have been explored in previous works and among other works we even cite the publications the reviewer mentions. Here our focus was on taking the view of representation learning and to define what is an invariant representation.
>
> > The difference between how the world works and how humans perceive is the central debate between causal and anti-causal. It is not whether the task is image classification or not. The arguments such as the work like IRM is only about causal but not anti-causal are plainly wrong.
>
> We are totally aware of such debates and did not intend to make such an argument in our writing. We are here retaking an argument that is often made, such as in [4]. The goal of our work was to think about what invariance can mean when the distribution of Y is stable across domains and no predictors are children of Y.
>
> > The algorithm is reframing the IRM algorithm
>
> We strongly disagree with this highly objectionable statement. We honestly fail to see how our algorithm and the one from IRM could be considered to be interchangeable, and we would appreciate if the reviewer could elaborate on this. If it was the case, other models, such as CORAL or MMD, could also be considered to be reframings of IRM, simply because they consist of an objective and a constraining penalty. In any case, the clear dissimilarity in empirical performance between our models and IRM should be enough for our contribution to not be dismissed in such a manner.
>
> Our approach outperforms 14 different baselines in different applications from fairness to domain generalization. While styles of writing can always be improved, we would appreciate it if the review could likewise be aligned with the empirical performance of the proposed algorithm.
>
> [1] Kim, Hyunjik, and Andriy Mnih. "Disentangling by factorising." International Conference on Machine Learning. PMLR, 2018.
>
> [2] Gondal, Muhammad Waleed, et al. "On the transfer of inductive bias from simulation to the real world: a new disentanglement dataset." NeurIPS (2019).
>
> [3] Wang, Yixin, and Michael I. Jordan. "Desiderata for representation learning: A causal perspective." arXiv preprint arXiv:2109.03795 (2021).
>
> [4] Heinze-Deml, Christina, and Nicolai Meinshausen. "Conditional variance penalties and domain shift robustness." Machine Learning (2021).

---

### Decision · Program_Chairs · 2022-01-20

**Decision:**

Reject

**Comment:**

The reviewers are in consensus. I recommend that the authors take their recommendations into consideration in revising their manuscript.